# Calcineurin inhibition enhances *Caenorhabditis elegans* lifespan by defecation defects-mediated calorie restriction and nuclear hormone signaling

**Priyanka Das[1], Alejandro Aballay[2], Jogender Singh[1]\***

[1]Department of Biological Sciences, Indian Institute of Science Education and Research, Mohali, India; [2]Department of Genetics, The University of Texas MD Anderson Cancer Center, Houston, United States

## eLife Assessment

This **important** study reveals insights into how calcineurin influences *C. elegans* pathogen susceptibility and lifespan through its role in controlling the defecation motor program. The authors provide **convincing** evidence to support a new mechanism through which calcineurin impacts longevity. This work will be of interest to investigators studying host-pathogen interactions and aging in a number of experimental systems.

**\*For correspondence:**
jogender@iisermohali.ac.in

**Competing interest:** The authors declare that no competing interests exist.

**Abstract** Calcineurin is a highly conserved calcium/calmodulin-dependent serine/threonine protein phosphatase with diverse functions. Inhibition of calcineurin is known to enhance the lifespan of *Caenorhabditis elegans* through multiple signaling pathways. Aiming to study the role of calcineurin in regulating innate immunity, we discover that calcineurin is required for the rhythmic defecation motor program (DMP) in *C. elegans*. Calcineurin inhibition leads to defects in the DMP, resulting in intestinal bloating, rapid colonization of the gut by bacteria, and increased susceptibility to bacterial infection. We demonstrate that intestinal bloating caused by calcineurin inhibition mimics the effects of calorie restriction, resulting in enhanced lifespan. The TFEB ortholog, HLH-30, is required for lifespan extension mediated by calcineurin inhibition. Finally, we show that the nuclear hormone receptor, NHR-8, is upregulated by calcineurin inhibition and is necessary for the increased lifespan. Our studies uncover a role for calcineurin in the *C. elegans* DMP and provide a new mechanism for calcineurin inhibition-mediated longevity extension.

## Introduction

Interventions that enhance lifespan also impart resistance to multiple stresses (*Johnson et al., 2001*). Indeed, the positive correlation between improved stress resistance and enhanced lifespan has been exploited to identify long-lived mutants (*de Castro et al., 2004*; *Denzel et al., 2014*; *Johnson et al., 2001*; *Muñoz and Riddle, 2003*; *Wang et al., 2004*). Among stress responses, innate immunity appears to be a crucial factor for enhanced lifespan (*Campos et al., 2021*; *Fabian et al., 2021*; *Soo et al., 2023*; *Xia et al., 2019*). However, the correlation between innate immunity and lifespan is not always positive. Interventions that alter lifespan may not modulate innate immunity, and vice-versa (*Labed et al., 2018*; *Naim et al., 2021*; *Otarigho and Aballay, 2020*; *Sun et al., 2011*). Some signaling pathways also establish a tradeoff between innate immunity and lifespan. Mutants that have improved immunity but reduced lifespan have been identified (*Amrit et al., 2019*; *Otarigho and*

**eLife digest** Many research efforts currently focus on identifying the dietary, pharmacological or genetic interventions that could help to prolong life. In the process, these investigations often uncover complex or even unexpected relationships between a range of physiological processes. The link between longevity and the immune system, for example, is yet to be fully understood.

To explore these dynamics, Das et al. focused on calcineurin, an enzyme present in organisms across the tree of life. In humans, calcineurin is known to regulate a set of proteins essential for the immune response; these proteins are absent in the microscopic worm *Caenorhabditis elegans*, in which inhibiting calcineurin extends lifespan. Investigating how calcineurin inhibition impacts the immune system of *C. elegans* therefore presents a unique opportunity to better understand the complex links between immunity and longevity.

Experiments conducted on worms genetically modified to lack calcineurin showed that these animals lived longer than their 'normal' counterparts, but that they were also more susceptible to infection when exposed to a harmful species of bacteria. Further experiments showed that the enzyme was crucial for regulating defecation in *C. elegans*. Without calcineurin, the worms became bloated and constipated; they could not properly eliminate bacteria, which could then proliferate in the digestive system and cause issues. However, intestinal bloating also activated signalling pathways normally triggered by calorie restriction – an intervention well-known for extending the lifespan of various species.

Taken together, the findings by Das et al. help explain why calcineurin inhibition in *C. elegans* leads to opposite effects on longevity and resistance to infection. They also align with a recent body of work showing the profound effect of gut bloating on food-seeking behaviors, immunity and lifespan. Further investigations into these mechanisms may one day uncover new ways to improve human health.

*Aballay, 2021*; *Ren and Ambros, 2015*). Conversely, mutants with enhanced lifespans but declined immune responses have also been discovered (*Kawli et al., 2010*). Moreover, the genetic pathways for lifespan and immunity could be uncoupled in mutants exhibiting enhanced lifespan and improved immune responses (*Alper et al., 2010*; *Guerrero et al., 2021*). Therefore, the relationship between lifespan and innate immunity appears to be complex and remains to be fully understood.

Calcineurin, a conserved protein from yeast to humans, is a calcium/calmodulin-dependent serine/threonine protein phosphatase that is involved in diverse cellular processes and signal transduction pathways (*Chen et al., 2022*; *Hogan et al., 2003*; *Schulz and Yutzey, 2004*; *Ulengin-Talkish and Cyert, 2023*). Dephosphorylation of substrate proteins by calcineurin affects several cellular pathways, including transcriptional signaling programs (*Ulengin-Talkish and Cyert, 2023*). Calcineurin regulates the activity of the transcription factors of the nuclear factor of activated T cells (NFAT) family (*Hogan et al., 2003*). Dephosphorylation of NFATs by calcineurin triggers their nuclear localization and activates immune responses in vertebrates (*Herbst et al., 2015*; *Hogan et al., 2003*; *Vandewalle et al., 2014*). In the nematode *Caenorhabditis elegans*, calcineurin regulates thermotaxis, body size, fertility, and lifespan (*Bandyopadhyay et al., 2002*; *Dong et al., 2007*; *Kuhara et al., 2002*; *Lee et al., 2013*). Knockdown of the catalytic subunit of calcineurin, *tax-6*, is known to enhance *C. elegans* lifespan via multiple pathways, including autophagy and CREB-regulated transcriptional coactivators (CRTCs; *Dong et al., 2007*; *Dwivedi et al., 2009*; *Mair et al., 2011*; *Tao et al., 2013*). However, the role of calcineurin in regulating *C. elegans* response to pathogen infections has not been studied. Because *C. elegans* lacks the NFAT transcription factors (*Song et al., 2013*), it will be intriguing to study how calcineurin inhibition impacts *C. elegans* innate immunity. These studies could also shed some light on the complex interplay between lifespan and immunity.

In this study, we examined the effect of calcineurin inhibition on *C. elegans* innate immunity. Surprisingly, we found that the knockdown of *tax-6* enhanced the susceptibility of *C. elegans* to bacterial infection despite enhancing lifespan. We discovered that *tax-6* is required for the rhythmic defecation motor program (DMP). The knockdown of *tax-6* resulted in intestinal bloating due to defects in the DMP, which enhanced susceptibility to bacterial infection by increasing gut colonization by bacteria. Intestinal bloating resulted in calorie restriction-like phenotypes, including reduced lipid

levels, and led to increased lifespan. We discovered that the TFEB ortholog, HLH-30, is required for calcineurin inhibition-mediated lifespan extension. Moreover, we found that the nuclear hormone receptor, NHR-8, is upregulated by calcineurin inhibition and is necessary for increased lifespan. Our studies uncover a new mechanism for calcineurin inhibition-mediated longevity extension.

## Results

### Calcineurin knockdown enhances *C. elegans* susceptibility to *Pseudomonas aeruginosa* infection

To understand the role of calcineurin in the innate immune response of *C. elegans*, we examined the survival of a hypomorphic allele of *tax-6*, *tax-6(p675)*, when exposed to the pathogenic bacterium *P. aeruginosa* PA14. Surprisingly, *tax-6(p675)* mutants showed significantly reduced survival on *P. aeruginosa* compared to wild-type N2 animals (*Figure 1A*). Similarly, animals carrying a null allele of *tax-6*, *tax-6(ok2065)*, also exhibited a significant decrease in survival on *P. aeruginosa* compared to N2 animals (*Figure 1B*). In addition, knockdown of *tax-6* by RNA interference (RNAi) increased susceptibility to *P. aeruginosa* relative to control animals (*Figure 1C*). The RNAi effects were specific to *tax-6*, as *tax-6(p675)* mutants did not show further increased susceptibility to *P. aeruginosa* when subjected to *tax-6* RNAi (*Figure 1D*). As reported earlier (*Dong et al., 2007*; *Dwivedi et al., 2009*; *Mair et al., 2011*; *Tao et al., 2013*), we confirmed that both *tax-6* loss of function and RNAi knockdown led to an increased lifespan on *E. coli* (*Figure 1E and F*). An earlier study demonstrated that calcineurin regulates cAMP response element-binding protein (CREB) and CRTCs to regulate lifespan in *C. elegans* (*Mair et al., 2011*). The knockdown of the CREB homolog-1 (*crh-1*) and *crtc-1* enhanced lifespan similar to the *tax-6* knockdown (*Mair et al., 2011*). We investigated whether *crh-1* and *crtc-1* knockdowns also affected *C. elegans* survival on *P. aeruginosa* as severely as *tax-6* knockdown. Interestingly, *crh-1* and *crtc-1* knockdowns did not compromise survival on *P. aeruginosa* as much as *tax-6* knockdown did (*Figure 1—figure supplement 1*). These findings suggested that, despite the increased lifespan, *tax-6* knockdown animals exhibit markedly enhanced susceptibility to *P. aeruginosa* infection through a mechanism likely independent of *crh-1* and *crtc-1*.

We observed enhanced matricidal hatching in *tax-6* knockdown animals on *P. aeruginosa*. Therefore, we asked whether the enhanced susceptibility of *tax-6* knockdown animals to *P. aeruginosa* was because of enhanced matricidal hatching. To this end, we studied the effects of *tax-6* knockdown in *fer-1(b232)* temperature-sensitive mutants. When grown at 25 °C, *fer-1(b232)* animals have unfertilized oocytes (*Argon and Ward, 1980*), thus eliminating the possibility of matricidal hatching. As shown in *Figure 1G*, even in the absence of matricidal hatching, *fer-1(b232)* animals exhibited reduced survival on *P. aeruginosa* following *tax-6* knockdown, indicating that the increased susceptibility is not due to enhanced matricidal hatching. Moreover, *fer-1(b232)* animals showed an extended lifespan on *E. coli* when *tax-6* was knocked down (*Figure 1H*). These results suggested that *tax-6* knockdown leads to both increased lifespan and enhanced susceptibility to pathogen infection.

We then explored whether the increased susceptibility to bacterial infection following *tax-6* knockdown was mediated by any known *C. elegans* innate immunity pathways, including the MAP kinase pathway mediated by NSY-1/SEK-1/PMK-1 (*Kim et al., 2002*), the MLK-1/MEK-1/KGB-1 c-Jun kinase pathway (*Kim et al., 2004*), the TGF-β/DBL-1 pathway (*Mallo et al., 2002*), and the bZIP transcription factor ZIP-2 pathway (*Estes et al., 2010*). In mutants deficient in each of these pathways, *tax-6* knockdown further increased susceptibility to *P. aeruginosa* (*Figure 1—figure supplement 2A–D*). The knockdown of *tax-6* appeared to have a more pronounced effect in *pmk-1(km25)* mutants than in other mutants, suggesting that inhibition of *tax-6* might exacerbate the adverse effects observed in *pmk-1(km25)* mutants. Additionally, the increased susceptibility to *P. aeruginosa* seen in *tax-6* knockdown animals was independent of the FOXO transcription factor DAF-16 (*Figure 1—figure supplement 2E*). Thus, the enhanced susceptibility to *P. aeruginosa* resulting from *tax-6* knockdown appears to operate independently of established *C. elegans* innate immunity pathways.

### Calcineurin is required for the *C. elegans* defecation motor program (DMP)

To understand why *tax-6* knockdown animals had enhanced susceptibility to *P. aeruginosa* infection, we studied the colonization of the intestine of *tax-6* RNAi animals by *P. aeruginosa*. Knockdown

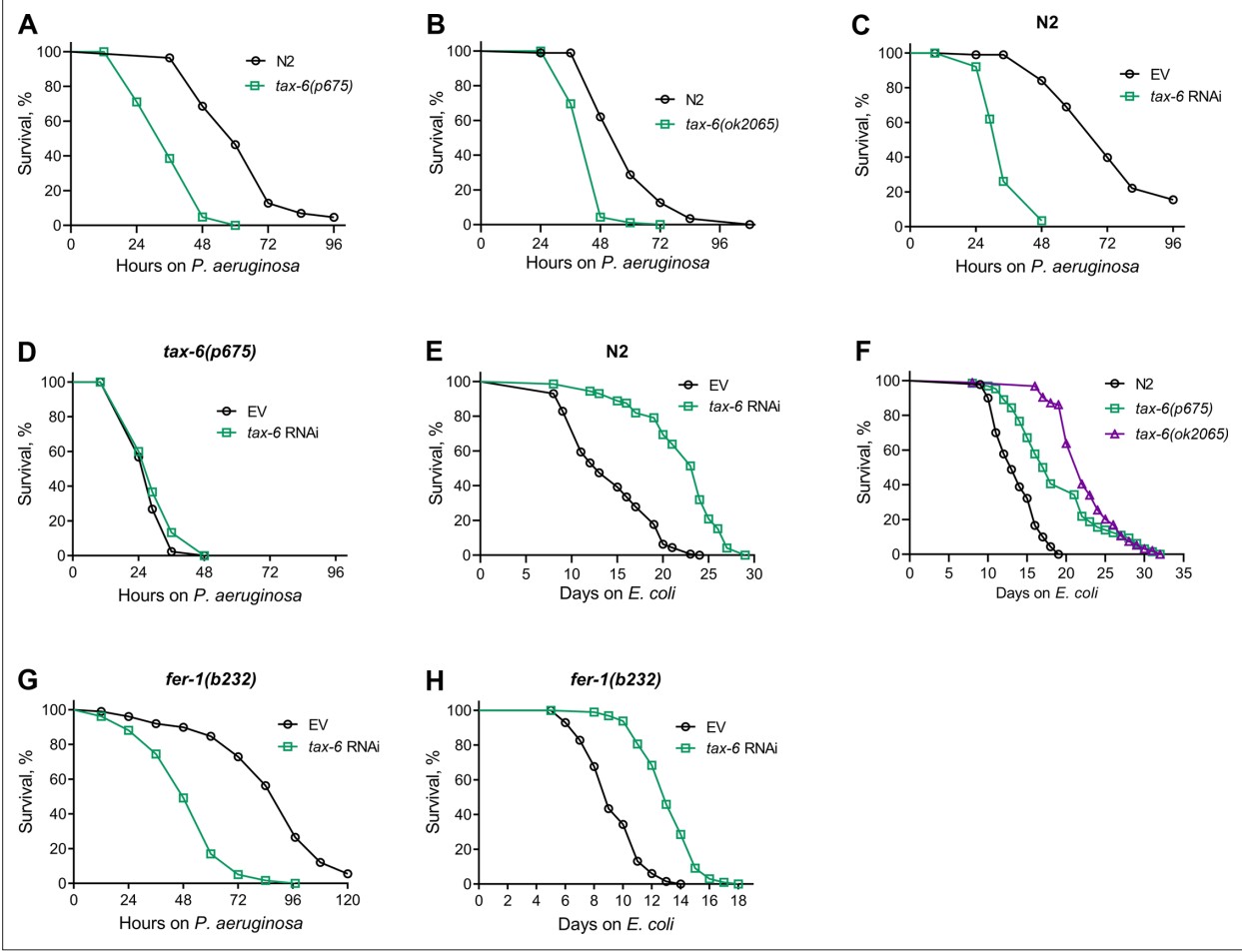

**Figure 1.** Calcineurin knockdown enhances *C. elegans* susceptibility to *P. aeruginosa*. (**A**) Representative survival plots of N2 and *tax-6(p675)* animals on *P. aeruginosa* PA14 at 25 °C. The animals were grown on *E. coli* OP50 at 20 °C until 1-day-old adults before transferring to *P. aeruginosa* PA14 at 25 °C. p<0.001. (**B**) Representative survival plots of N2 and *tax-6(ok2065)* animals on *P. aeruginosa* PA14 at 25 °C. The animals were grown on *E. coli* OP50 at 20 °C until 1-day-old adults before transferring to *P. aeruginosa* PA14 at 25 °C. p<0.001. (**C**) Representative survival plots of N2 animals on *P. aeruginosa* PA14 at 25 °C after treatment with the empty vector (EV) control and *tax-6* RNAi. p<0.001. (**D**) Representative survival plots of *tax-6(p675)* animals on *P. aeruginosa* PA14 at 25 °C after treatment with the EV control and *tax-6* RNAi. n.s., nonsignificant. (**E**) Representative survival plots of N2 animals grown on bacteria for RNAi against *tax-6* along with the EV control at 20 °C. Day 0 represents young adults. p<0.001. (**F**) Representative survival plots of N2, *tax-6(p675)*, and *tax-6(ok2065)* animals grown on *E. coli* OP50 at 20 °C. Day 0 represents young adults. p<0.001 for *tax-6(p675)* and *tax-6(ok2065)* compared to N2. (**G**) Representative survival plots of *fer-1(b232)* animals on *P. aeruginosa* PA14 at 25 °C after treatment with the EV control and *tax-6* RNAi at 25 °C. p<0.001. (**H**) Representative survival plots of *fer-1(b232)* animals grown on bacteria for RNAi against *tax-6* along with the EV control at 25 °C. The animals were developed at 25 °C. Day 0 represents young adults. p<0.001. For all panels, n=3 biological replicates; animals per condition per replicate >60.

The online version of this article includes the following source data and figure supplement(s) for figure 1:

**Source data 1.** Calcineurin knockdown enhances *C. elegans* susceptibility to *P. aeruginosa*.

**Figure supplement 1.** Calcineurin knockdown enhances *C. elegans* susceptibility to *P. aeruginosa*.

**Figure supplement 1—source data 1.** Calcineurin knockdown enhances *C. elegans* susceptibility to *P. aeruginosa*.

**Figure supplement 2.** Calcineurin inhibition enhances susceptibility to *P. aeruginosa* independent of known immunity pathways.

**Figure supplement 2—source data 1.** Calcineurin inhibition enhances susceptibility to *P. aeruginosa* independent of known immunity pathways.

of *tax-6* led to enhanced intestinal colonization by *P. aeruginosa* compared to the control animals (*Figure 2A and B*). Similarly, *tax-6(ok2065)* and *tax-6(p675)* animals also showed increased intestinal colonization by *P. aeruginosa* compared to N2 animals (*Figure 2C and D* and *Figure 2—figure supplement 1A, B*). Animals with bloated intestinal lumens exhibit enhanced bacterial colonization of the gut, an improved lifespan, and an increased susceptibility to pathogens (*Kumar et al., 2019*;

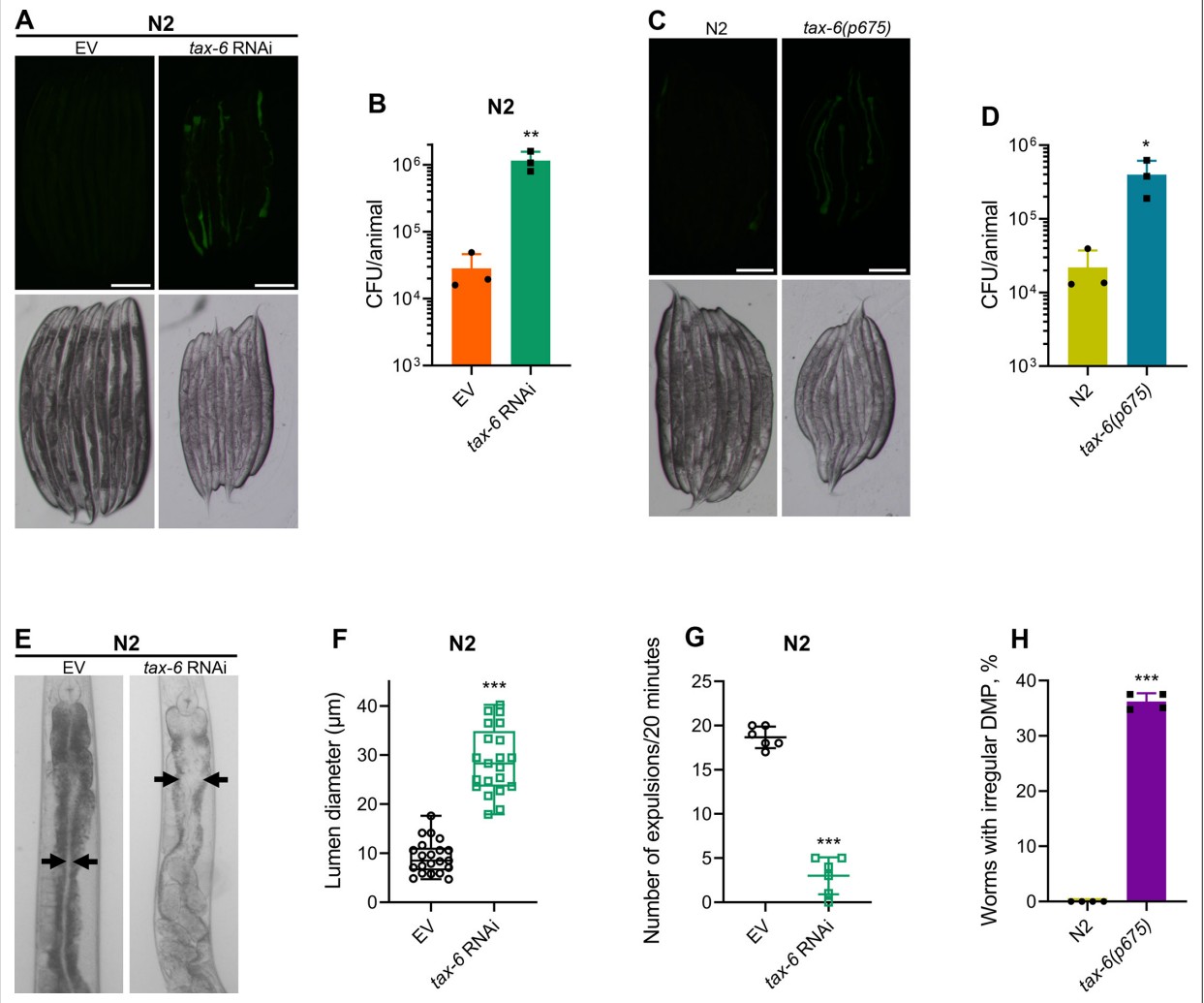

**Figure 2.** Calcineurin is required for the *C. elegans* defecation motor program (DMP). (**A**) Representative fluorescence (top) and the corresponding brightfield (bottom) images of N2 animals incubated on *P. aeruginosa*-GFP for 6 hr at 25 °C after growth on the empty vector (EV) control and *tax-6* RNAi bacteria. Scale bar=200 µm. (**B**) Colony-forming units (CFU) per animal of N2 worms incubated on *P. aeruginosa*-GFP for 6 hr at 25 °C after growth on the EV control and *tax-6* RNAi bacteria. **p<0.01 via the *t*-test (n=3 biological replicates). (**C**) Representative fluorescence (top) and the corresponding brightfield (bottom) images of N2 and *tax-6(p675)* animals incubated on *P. aeruginosa*-GFP for 6 hours at 25 °C after growth on *E. coli* OP50 at 20 °C. Scale bar=200 µm. (**D**) CFU per animal of N2 and *tax-6(p675)* worms incubated on *P. aeruginosa*-GFP for 6 hours at 25 °C after growth on *E. coli* OP50 at 20 °C. *p<0.05 via the *t*-test (n=3 biological replicates). (**E**) Representative photomicrographs of N2 animals grown on the EV control and *tax-6* RNAi bacteria at 20 °C until 1-day-old adults. Arrows point to the border of the intestinal lumen. (**F**) Quantification of the diameter of the intestinal lumen of N2 animals grown on the EV control and *tax-6* RNAi bacteria at 20 °C until 1-day-old adults. ***p<0.001 via the *t*-test (n=21 worms each). (**G**) The number of expulsion events observed in 20 min in 1-day-old adult N2 animals grown on the EV control and *tax-6* RNAi bacteria at 20 °C. ***p<0.001 via the *t*-test (n=6 worms each). (**H**) Percent of 1-day-old adult worms having irregular DMP. ***p<0.001 via the *t*-test (n=4 biological replicates).

The online version of this article includes the following source data and figure supplement(s) for figure 2:

**Source data 1.** Calcineurin is required for the *C. elegans* defecation motor program.

**Figure supplement 1.** Calcineurin is required for the *C. elegans* defecation motor program (DMP).

**Figure supplement 1—source data 1.** Calcineurin is required for the *C. elegans* defecation motor program.

**Figure supplement 2.** Calcineurin inhibition leads to defects in the defecation motor program (DMP) without affecting the pharyngeal pumping rate.

**Figure supplement 2—source data 1.** Calcineurin inhibition leads to defects in the defecation motor program without affecting the pharyngeal pumping rate.

*Singh and Aballay, 2019a*). Because *tax-6* knockdown animals displayed all of these phenotypes, we asked whether *tax-6* knockdown led to the bloating of the intestinal lumen. Indeed, we found that *tax-6* knockdown resulted in bloated intestinal lumens (*Figure 2E and F*).

Intestinal bloating could result from either defects in pharyngeal pumping (*Kumar et al., 2019*) or defects in the defecation motor program (DMP) (*Singh and Aballay, 2019b*). We found that *tax-6* knockdown did not affect the pharyngeal pumping rate (*Figure 2—figure supplement 2A*). This prompted us to explore whether calcineurin is necessary for the DMP. The *C. elegans* DMP is a highly coordinated rhythmic behavior required for the regular expulsion of intestinal contents, occurring approximately once per minute. The DMP involves a series of muscle contractions, including posterior body muscle contraction (pBoc), anterior body muscle contraction (aBoc), and expulsion muscle contraction (EMC), to release intestinal contents (*Thomas, 1990*). When we counted expulsion events over a 20-min period, we found that *tax-6* knockdown animals had a drastically reduced number of expulsion events (*Figure 2G*). The DMP-defect phenotype had low penetrance in *tax-6(p675)* mutants, which exhibited both regular and irregular DMPs (*Figure 2—figure supplement 2B*). At the 1-day-old adult stage, about 36% of *tax-6(p675)* animals showed irregular and slowed DMP, while the remainder had regular DMP (*Figure 2H*), suggesting that *tax-6(p675)* is a weak allele. The fraction of the animals with irregular DMP appeared to increase with age, indicating that this phenotype might be age-dependent. This may also explain why *tax-6(p675)* animals were reported to have a normal defecation cycle in an earlier study (*Lee et al., 2005*).

Next, we investigated whether the reduced number of expulsion events was due to regular intervals with longer cycle lengths or if rhythmicity was entirely disrupted upon *tax-6* knockdown. To assess this, we obtained ethograms of the DMP for N2 animals grown on control and *tax-6* RNAi. While animals on control RNAi displayed regular cycles of pBoc, aBoc, and EMC, the *tax-6* RNAi animals exhibited disrupted rhythmicity (*Figure 3A* and *Figure 3—figure supplement 1*). Most *tax-6* knockdown animals lacked the pBoc and aBoc steps and had sporadic expulsion events. Isolated pBoc events were occasionally observed, indicating a complete loss of rhythmicity in *tax-6* knockdown animals. Ethograms for *tax-6(ok2065)* animals also showed disrupted rhythmicity (*Figure 3B* and *Figure 3—figure supplement 2*). Although the number of expulsion events appeared higher in *tax-6(ok2065)* animals compared to *tax-6* RNAi animals (*Figure 3—figure supplements 1 and 2*), these expulsion events seemed superficial, releasing little to no gut content. This suggested slow movement of gut content in *tax-6(ok2065)* animals, leading to constipation and intestinal bloating. We examined gut content movement by measuring the clearance of blue dye (erioglaucine disodium salt) from the gut. The clearance was significantly slower in *tax-6(ok2065)* animals compared to N2 animals (*Figure 3C*), indicating impaired gut content movement due to the loss of *tax-6*. Similarly, *tax-6* knockdown animals also showed significantly slowed gut content movement (*Figure 3D*).

We then explored whether the disruption of DMP rhythmicity due to *tax-6* knockdown affected *P. aeruginosa* responses similarly to longer but regular DMP cycles. To do this, we studied *P. aeruginosa* colonization in *clk-1(qm30)* and *isp-1(qm150)* mutants, which have regular but extended DMP cycles (*Feng et al., 2001*; *Wong et al., 1995*). Interestingly, both *clk-1(qm30)* and *isp-1(qm150)* mutants showed significantly reduced intestinal colonization by *P. aeruginosa* compared to N2 animals (*Figure 3—figure supplement 3A–D*). This reduced colonization could be attributed to their significantly decreased pharyngeal pumping rates (*Wong et al., 1995*; *Yee et al., 2014*), suggesting a lower intake of bacterial food in these mutants. While the survival of *clk-1(qm30)* animals on *P. aeruginosa* was comparable to N2 animals (*Figure 3—figure supplement 3E*), *isp-1(qm150)* animals exhibited significantly improved survival (*Figure 3—figure supplement 3F*). Conversely, knockdown of *flr-1*, *nhx-2*, and *pbo-1* in N2 animals resulted in significantly reduced survival on *P. aeruginosa* compared to control RNAi (*Figure 3—figure supplement 3G*). Knockdown of these genes causes complete disruption of DMP rhythmicity, increasing gut colonization by *P. aeruginosa* (*Singh and Aballay, 2019b*). Overall, these findings demonstrated that calcineurin is crucial for maintaining the DMP ultradian clock, and its inhibition increases susceptibility to *P. aeruginosa* by disrupting the DMP.

## Calcineurin inhibition enhances lifespan via DMP defects-induced calorie restriction

Defects in the *C. elegans* DMP lead to reduced nutrient absorption, resulting in decreased intestinal fat levels (*Sheng et al., 2015*). We investigated whether *tax-6* knockdown similarly reduces fat levels.

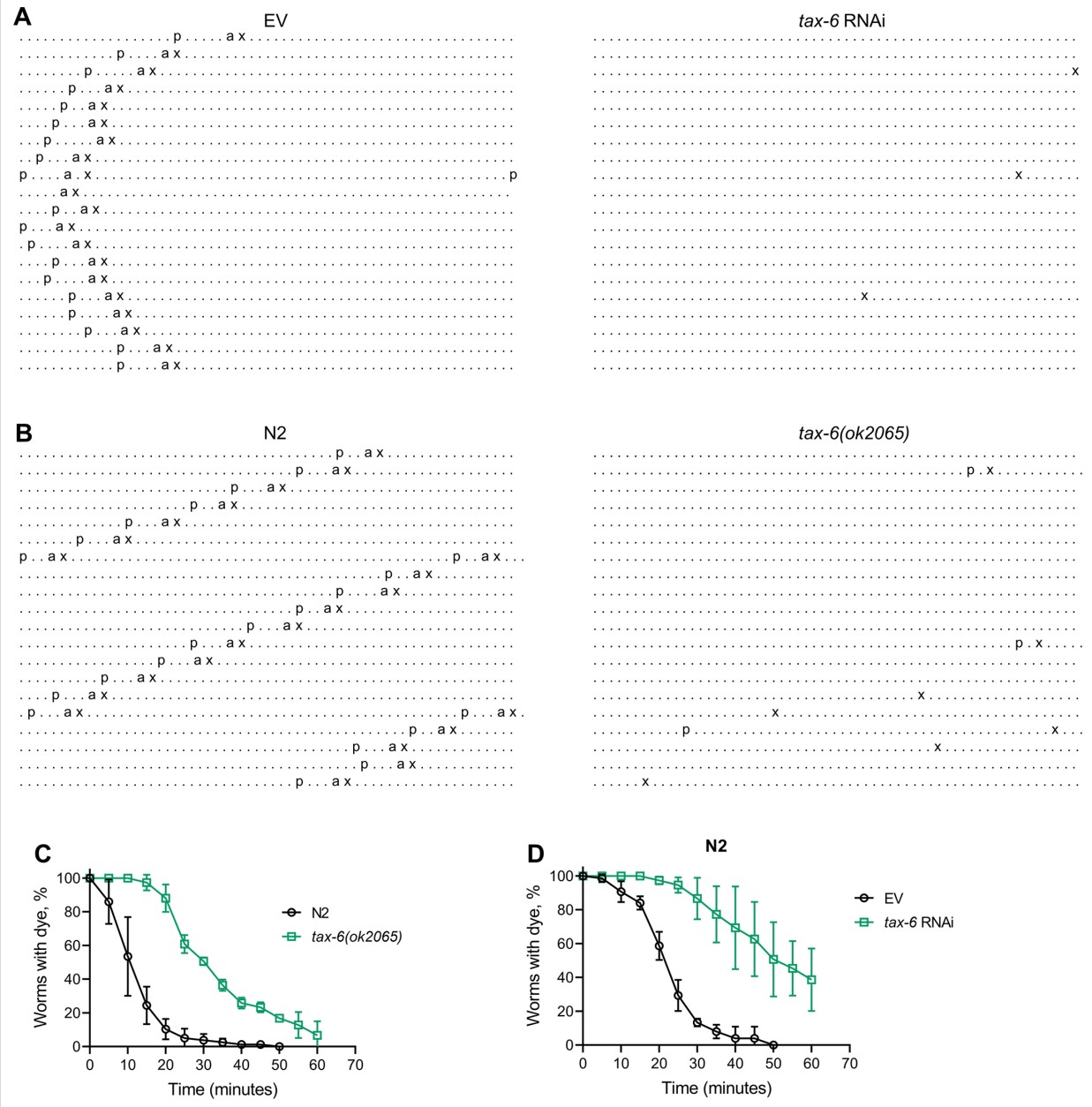

**Figure 3.** Calcineurin inhibition disrupts the *C. elegans* defecation motor program (DMP). (**A**) Representative DMP ethograms of N2 animals after growth on the empty vector (EV) control and *tax-6* RNAi bacteria till 1-day-old stage. Each dot represents a second, and each row represents a minute. 'p', 'a', and 'x' represent posterior body-wall muscle contraction, anterior body-wall muscle contraction, and expulsion muscle contraction, respectively. (**B**) Representative DMP ethograms of N2 and *tax-6(ok2065)* animals after their growth on *E. coli* OP50 at 20 °C till the 1-day-old adult stage. (**C**) Plots of the change in the percent of animals with blue dye in their gut with time for 1-day-old adult N2 and *tax-6(ok2065)* animals (n=3 biological replicates). (**D**) Plots of the change in the percent of animals with blue dye in their gut with time for 1-day-old adult N2 animals grown on EV control and *tax-6* RNAi bacteria (n=3 biological replicates).

The online version of this article includes the following source data and figure supplement(s) for figure 3:

**Source data 1.** Calcineurin inhibition disrupts the *C. elegans* defecation motor program.

**Figure supplement 1.** Calcineurin inhibition disrupts the *C. elegans* defecation motor program (DMP).

**Figure supplement 2.** Calcineurin inhibition disrupts the *C. elegans* defecation motor program (DMP).

**Figure supplement 3.** Slow DMP versus disrupted DMP lead to distinct phenotypes on *P. aeruginosa* exposure.

**Figure supplement 3—source data 1.** Slow DMP versus disrupted DMP lead to distinct phenotypes on *P. aeruginosa* exposure.

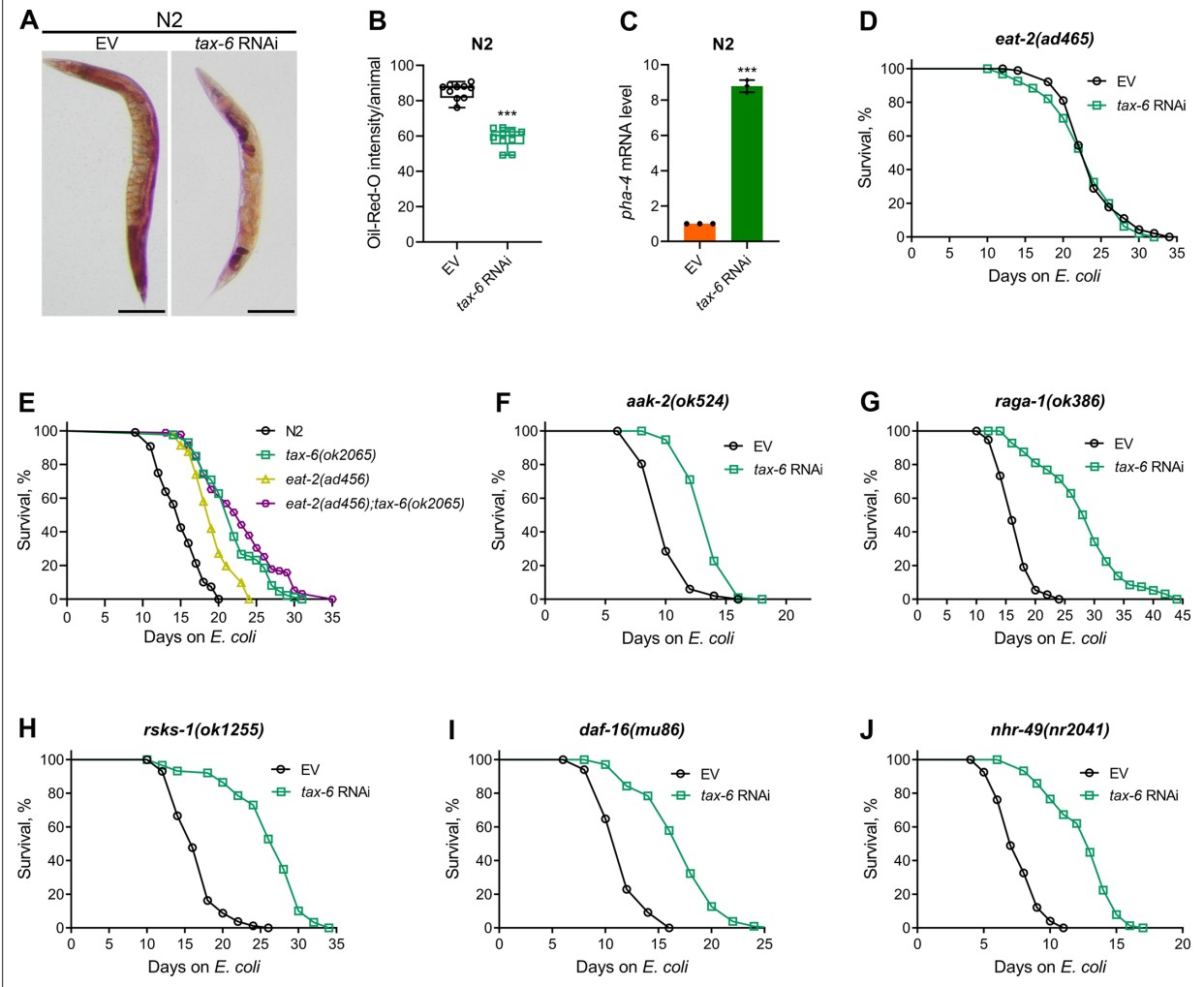

**Figure 4.** Calcineurin knockdown enhances lifespan via DMP defects-mediated calorie restriction. (**A**) Representative photomicrographs of oil-red-O (ORO) stained 1-day-old adult N2 animals grown on the empty vector (EV) control and *tax-6* RNAi bacteria at 20 °C. Scale bar=200 µm. (**B**) Quantification of ORO intensity per animal of 1-day-old adult N2 animals grown on the EV control and *tax-6* RNAi bacteria at 20 °C. The values are area normalized for each animal. ***p<0.001 via the *t*-test (n=10 worms each). (**C**) Quantitative reverse transcription-PCR (qRT-PCR) for *pha-4* mRNA levels in N2 animals grown on the EV control and *tax-6* RNAi bacteria at 20 °C until 1-day-old adults. ***p<0.001 (n=3 biological replicates). (**D**) Representative survival plots of *eat-2(ad465)* animals grown on the bacteria for RNAi against *tax-6* along with the EV control at 20 °C. Day 0 represents young adults. n.s., nonsignificant (n=3 biological replicates; animals per condition per replicate >80). (**E**) Representative survival plots of N2, *tax-6(ok2065)*, *eat-2(ad465)*, and *eat-2(ad465);tax-6(ok2065)* animals grown on *E. coli* OP50 at 20 °C. Day 0 represents young adults. p<0.001 for *tax-6(ok2065)*, *eat-2(ad465)*, and *eat-2(ad465);tax-6(ok2065)* compared to N2. p<0.05 for *eat-2(ad465);tax-6(ok2065)* compared to *tax-6(ok2065)* (n=3 biological replicates; animals per condition per replicate >65). (**F–J**) Representative survival plots of *aak-2(ok524)* (**E**), *raga-1(ok386)* (**F**), *rsks-1(ok1255)* (**G**), *daf-16(mu86)* (**H**), and *nhr-49(nr2041)* (**I**) animals grown on the bacteria for RNAi against *tax-6* along with the EV control at 20 °C. Day 0 represents young adults. p<0.001 for all the plots (n=3 biological replicates; animals per condition per replicate >60).

The online version of this article includes the following source data and figure supplement(s) for figure 4:

**Source data 1.** Calcineurin knockdown enhances lifespan via DMP defects-mediated calorie restriction.

**Figure supplement 1.** The *eat-2(ad465)* animals are not defective in RNAi.

To test this, we performed oil-red-O (ORO) staining on *tax-6* RNAi animals. The knockdown of *tax-6* significantly reduced fat levels (***Figure 4A and B***). Next, we examined whether the reduced fat levels in *tax-6* knockdown animals mimic calorie restriction-like phenotypes. Calorie restriction is known to upregulate the expression of the FoxA transcription factor PHA-4 (***Panowski et al., 2007***). Consistent with this, *tax-6* knockdown also resulted in the upregulation of *pha-4* expression levels (***Figure 4C***). Because calorie restriction enhances lifespan (***Kaeberlein et al., 2006***; ***Lakowski and Hekimi, 1998***; ***Panowski et al., 2007***), we explored whether calcineurin inhibition could similarly increase lifespan

through calorie restriction mechanisms. The knockdown of *tax-6* in the genetic model of calorie restriction, *eat-2(ad465)* (*Lakowski and Hekimi, 1998*), did not increase the lifespan (*Figure 4D*).

Since *eat-2(ad465)* animals consume fewer bacteria, it is possible that their feeding-based RNAi efficiency is reduced, leading to a lack of responsiveness to *tax-6* RNAi. To assess RNAi efficiency in *eat-2(ad465)* animals, we exposed them to *act-5* and *bli-3* RNAi, which are known to cause developmental arrest and a blistered cuticle, respectively (*Gahlot and Singh, 2024*). Similar to N2 animals, *eat-2(ad465)* mutants were sensitive to the toxic effects of *act-5* and *bli-3* knockdown (*Figure 4—figure supplement 1*), indicating that *eat-2(ad465)* animals are not defective in RNAi. To further confirm that calcineurin inhibition enhances lifespan via calorie restriction, we examined the lifespan of *eat-2(ad465);tax-6(ok2065)* double mutants. While *eat-2(ad465)* animals displayed increased lifespan compared to N2 controls, *eat-2(ad465);tax-6(ok2065)* double mutants had a lifespan similar to *tax-6(ok2065)* animals (*Figure 4E*). These results suggested that calcineurin inhibition promotes lifespan extension through calorie restriction mechanisms.

The mechanisms of lifespan enhancement by dietary or calorie restriction are complex and involve several different downstream pathways (*Chamoli et al., 2014*; *Greer and Brunet, 2009*; *Hansen et al., 2008*). Further, different pathways are operational under different dietary regimens (*Greer and Brunet, 2009*). Therefore, we investigated the involvement of pathways associated with calorie restriction-mediated lifespan extension in response to calcineurin inhibition. Specifically, we analyzed the interactions of the AMP-activated kinase, *aak-2* (*Greer and Brunet, 2009*), the mTOR pathway, *raga-1* (*Robida-Stubbs et al., 2012*; *Zhang et al., 2019*), ribosomal S6 kinase, *rsks-1* (*Chen et al., 2009*; *Selman et al., 2009*; *Zhang et al., 2019*), the FOXO transcription factor, *daf-16* (*Greer and Brunet, 2009*), and the nuclear hormone receptor, *nhr-49* (*Chamoli et al., 2014*) with calcineurin inhibition in regulating lifespan. Mutants defective in *aak-2* appeared to have only a partial increase in lifespan upon calcineurin inhibition (*Figure 4F*). In contrast, loss-of-function mutants of *raga-1*, *rsks-1*, *daf-16*, and *nhr-49* exhibited enhanced lifespan upon calcineurin inhibition (*Figure 4G–J*).

## Calcineurin knockdown enhances lifespan via HLH-30 and NHR-8

To further investigate longevity pathways regulated by calorie restriction that may enhance lifespan downstream of calcineurin inhibition, we examined the role of the TFEB ortholog HLH-30. HLH-30 is essential for the starvation response and triggers lipid depletion under nutrient-deprived conditions (*O'Rourke and Ruvkun, 2013*). Knockdown of *tax-6* did not extend the lifespan of *hlh-30(tm1978)* animals (*Figure 5A*), indicating that HLH-30 is required for the increased lifespan observed with calcineurin inhibition. To determine whether HLH-30 is necessary for DMP defect-mediated lifespan extension, we assessed the lifespan of *hlh-30(tm1978)* animals upon knockdown of *flr-1*, *nhx-2*, and *pbo-1*. Knockdown of these genes did not alter the lifespan of *hlh-30(tm1978)* animals compared to control RNAi (*Figure 5A*), except for a slight increase in lifespan with *nhx-2* knockdown. The *hlh-30(tm1978)* animals were found to be sensitive to RNAi (*Figure 5—figure supplement 1*), suggesting that the lack of lifespan change with *tax-6*, *flr-1*, and *pbo-1* RNAi was not due to reduced RNAi efficiency in *hlh-30(tm1978)* animals. Additionally, we confirmed that *tax-6* knockdown caused DMP defects and intestinal bloating in *hlh-30(tm1978)* animals (*Figure 5—figure supplement 2A–C*). We also explored whether HLH-30 affected fat depletion due to DMP defects. The ORO levels declined significantly in *hlh-30(tm1978)* animals upon the knockdown of *tax-6* and other DMP-regulating genes (*Figure 5B and C*). The reduction in ORO levels was less pronounced in *tax-6* knockdown animals compared to those with other DMP-regulating gene knockdowns.

Steroid hormone signaling mediated by NHR-8 is known to regulate lifespan extension under calorie restriction (*Thondamal et al., 2014*). We tested whether NHR-8 was necessary for lifespan extension induced by calcineurin inhibition. Indeed, *tax-6* knockdown did not extend the lifespan of *nhr-8(ok186)* animals (*Figure 5D*). While the specific ligands of NHR-8 are not well-defined (*Magner et al., 2013*), steroid hormones produced by the cytochrome P450 enzyme DAF-9 have been shown to enhance lifespan via NHR-8 under calorie restriction (*Thondamal et al., 2014*). Therefore, we investigated whether DAF-9 mediated the enhanced lifespan downstream of calcineurin inhibition. We found that *tax-6* knockdown enhanced the lifespan of *daf-9(rh50)* animals (*Figure 5E*), indicating that DAF-9-produced steroid hormones were not required for lifespan increment by calcineurin inhibition. We also examined the lifespan of *nhr-8(ok186)* animals upon knockdown of *flr-1*, *nhx-2*, and *pbo-1*. Knockdown of these genes did not affect the lifespan of *nhr-8(ok186)* animals compared to control

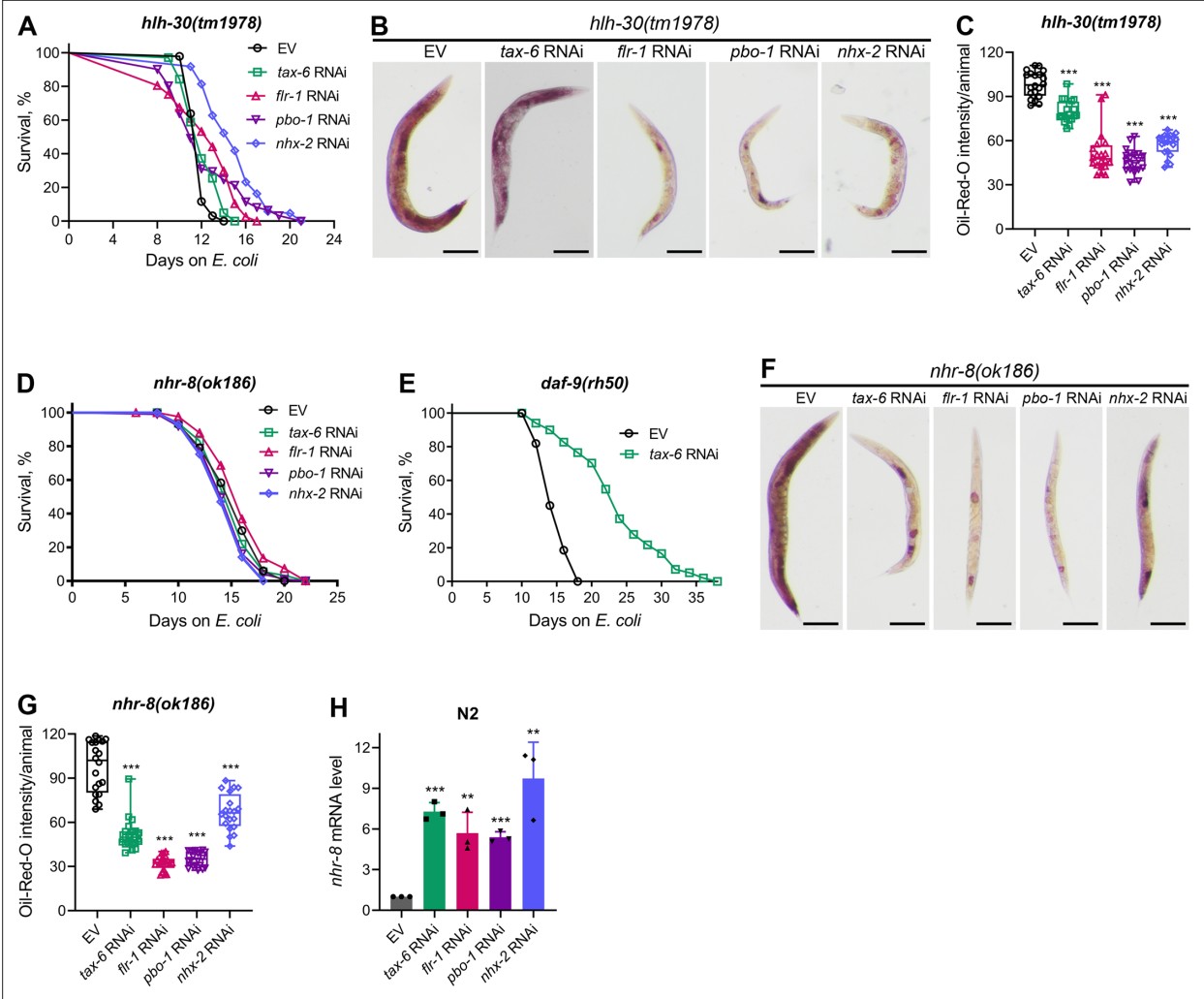

**Figure 5.** Calcineurin inhibition enhances lifespan via HLH-30 and NHR-8. (**A**) Representative survival plots of *hlh-30(tm1978)* animals grown on bacteria for RNAi against *tax-6*, *flr-1*, *pbo-1*, and *nhx-2* along with the empty vector (EV) control at 20 °C. Day 0 represents young adults. p values for *tax-6*, *flr-1*, *pbo-1*, and *nhx-2* compared to EV are <0.01,<0.001, nonsignificant, and <0.001, respectively (n=3 biological replicates; animals per condition per replicate >60). (**B**) Representative photomicrographs of oil-red-O (ORO) stained 1-day-old adult *hlh-30(tm1978)* animals grown on the EV control, *tax-6*, *flr-1*, *pbo-1*, and *nhx-2* RNAi bacteria at 20 °C. Scale bar=200 µm. (**C**) Quantification of ORO intensity per animal of 1-day-old adult *hlh-30(tm1978)* animals grown on the EV control, *tax-6*, *flr-1*, *pbo-1*, and *nhx-2* RNAi bacteria at 20 °C. The values are area normalized for each animal. ***p<0.001 via the *t*-test (n=20 worms each). (**D**) Representative survival plots of *nhr-8(ok186)* animals grown on bacteria for RNAi against *tax-6*, *flr-1*, *pbo-1*, and *nhx-2* along with the EV control at 20 °C. Day 0 represents young adults. p values for *tax-6*, *flr-1*, *pbo-1*, and *nhx-2* compared to EV are nonsignificant,<0.05, nonsignificant, and <0.05, respectively (n=3 biological replicates; animals per condition per replicate >70). (**E**) Representative survival plots of *daf-9(rh50)* animals grown on the bacteria for RNAi against *tax-6* along with the EV control at 20 °C. Day 0 represents young adults. p<0.001 (n=3 biological replicates; animals per condition per replicate >80). (**F**) Representative photomicrographs of ORO stained 1-day-old adult *nhr-8(ok186)* animals grown on the EV control, *tax-6*, *flr-1*, *pbo-1*, and *nhx-2* RNAi bacteria at 20 °C. Scale bar=200 µm. (**G**) Quantification of ORO intensity per animal of 1-day-old adult *nhr-8(ok186)* animals grown on the EV control, *tax-6*, *flr-1*, *pbo-1*, and *nhx-2* RNAi bacteria at 20 °C. The values are area normalized for each animal. ***p<0.001 via the *t*-test (n=20 worms each). (**H**) Quantitative reverse transcription-PCR (qRT-PCR) for *nhr-8* mRNA levels in N2 animals grown on the EV control, *tax-6*, *flr-1*, *pbo-1*, and *nhx-2* RNAi bacteria at 20 °C until 1-day-old adults. ***p<0.001 via the *t*-test (n=3 biological replicates).

The online version of this article includes the following source data and figure supplement(s) for figure 5:

**Source data 1.** Calcineurin inhibition enhances lifespan via HLH-30 and NHR-8.

**Figure supplement 1.** The *hlh-30(tm1978)* and *nhr-8(ok186)* animals are not defective in RNAi.

**Figure supplement 2.** Calcineurin inhibition leads to defects in the defecation motor program (DMP) in *hlh-30(tm1978)* and *nhr-8(ok186)* animals.

**Figure supplement 2—source data 1.** Calcineurin inhibition leads to defects in the defecation motor program in *hlh-30(tm1978)* and *nhr-8(ok186)* animals.

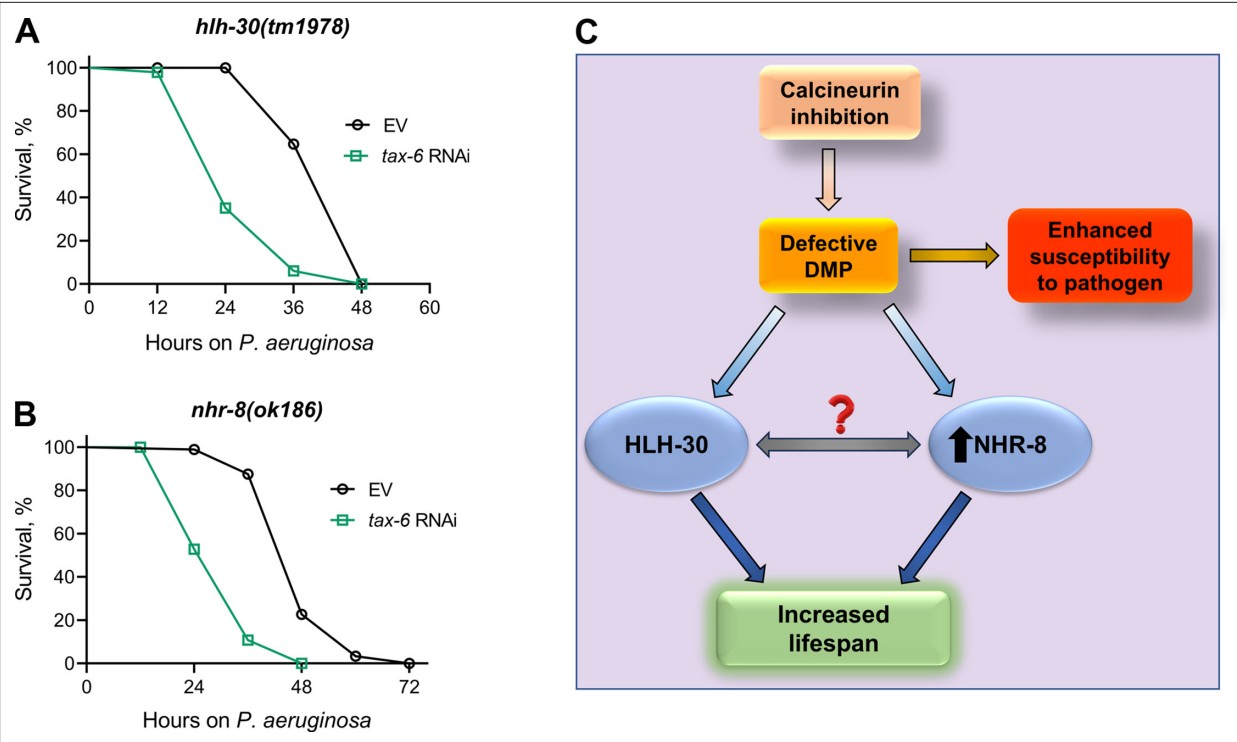

**Figure 6.** Calcineurin inhibition-mediated effects on lifespan and survival on *P. aeruginosa* are mediated by distinct mechanisms. (**A**)-(**B**) Representative survival plots of *hlh-30(tm1978)* (**A**) and *nhr-8(ok186)* (**B**) animals on *P. aeruginosa* PA14 at 25 °C after treatment with the empty vector (EV) control and *tax-6* RNAi. p<0.001 for both the plots (n=3 biological replicates; animals per condition per replicate >85). (**C**) Model depicting the mechanism of increased lifespan and enhanced susceptibility to pathogen upon inhibition of calcineurin via the defects in the defecation motor program (DMP).

The online version of this article includes the following source data for figure 6:

**Source data 1.** Calcineurin inhibition-mediated effects on lifespan and survival on *P. aeruginosa* are mediated by distinct mechanisms.

RNAi (*Figure 5D*). As with *hlh-30(tm1978)* animals, the lack of lifespan change in *nhr-8(ok186)* animals was not due to RNAi insensitivity, as they were responsive to RNAi (*Figure 5—figure supplement 1*). Furthermore, *tax-6* knockdown resulted in DMP defects in *nhr-8(ok186)* animals (*Figure 5—figure supplement 2D*). We investigated whether NHR-8 influenced fat depletion upon DMP defects. A significant reduction in fat levels was observed in *nhr-8(ok186)* animals upon knockdown of *tax-6*, *flr-1*, *nhx-2*, and *pbo-1* (*Figure 5F and G*), suggesting that NHR-8 functions downstream of or in parallel with fat depletion.

Since calorie restriction can enhance the expression of genes that promote lifespan, such as *pha-4* (*Panowski et al., 2007*), we examined whether *tax-6* knockdown modulates *nhr-8* mRNA levels. We observed a significant increase in *nhr-8* mRNA levels following *tax-6* knockdown (*Figure 5H*). To determine whether this upregulation was due to intestinal bloating, we assessed *nhr-8* mRNA levels in *flr-1*, *nhx-2*, and *pbo-1* knockdown animals. Knockdown of these genes also increased *nhr-8* mRNA levels (*Figure 5H*), indicating that intestinal bloating induces *nhr-8* upregulation. Thus, calcineurin inhibition likely enhances lifespan via NHR-8 by increasing its expression.

Given that HLH-30 and NHR-8 are essential for lifespan extension upon calcineurin inhibition, we investigated whether these pathways also influence survival in response to *P. aeruginosa* infection following calcineurin knockdown. Both *hlh-30(tm1978)* and *nhr-8(ok186)* animals showed significantly reduced survival upon *tax-6* RNAi (*Figure 6A and B*). These findings suggested that the reduced survival on *P. aeruginosa* following calcineurin inhibition is independent of HLH-30 and NHR-8 and is more likely due to increased gut colonization by *P. aeruginosa* resulting from DMP defects (*Figure 6C*).

## Discussion

We discovered that calcineurin is required for the *C. elegans* DMP. Calcineurin activity is regulated by intracellular calcium levels. Increased amounts of calcium ions activate the phosphatase activity of calcineurin via the binding of the calcium-sensing protein calmodulin to calcineurin (*Klee et al., 1998*). The rhythmic DMP cycle in *C. elegans* is known to be regulated by rhythmic calcium waves (*Dal Santo et al., 1999*; *Teramoto and Iwasaki, 2006*). The calcium waves are regulated by the endoplasmic reticulum calcium channel, ITR-1, and mutations in the *itr-1* gene affect defecation by preventing cytoplasmic calcium release (*Dal Santo et al., 1999*). It is likely that calcium waves regulate DMP via calcineurin activities. Indeed, calcineurin is expressed in the enteric muscles that are required for contractions for DMP (*Lee et al., 2005*). It is interesting to note that *tax-6* gain-of-function mutants are also known to have DMP defects (*Lee et al., 2005*). Therefore, optimum calcineurin activity appears to be crucial for maintaining a rhythmic DMP.

Calcineurin inhibition has been shown to extend *C. elegans* lifespan via multiple mechanisms (*Dong et al., 2007*; *Dwivedi et al., 2009*; *Mair et al., 2011*; *Tao et al., 2013*). Activation of autophagy has been shown to be one of the mechanisms of extended lifespan upon calcineurin inhibition (*Dwivedi et al., 2009*). We showed that defects in the DMP caused by calcineurin inhibition reduce lipid levels and mimic calorie restriction. Calorie restriction is known to induce autophagy (*Morselli et al., 2010*), and autophagy is required for calorie restriction-mediated lifespan extension (*Hansen et al., 2008*; *Jia and Levine, 2007*). Therefore, it is likely that reduced lipid levels because of intestinal bloating result in the activation of autophagy upon calcineurin inhibition. Importantly, the TFEB ortholog, HLH-30, is required for lipolysis and autophagy under starvation conditions (*Lapierre et al., 2013*; *O'Rourke and Ruvkun, 2013*). HLH-30 is also required for lifespan enhancement via autophagy and multiple longevity pathways (*Lapierre et al., 2013*; *O'Rourke and Ruvkun, 2013*). We observed that HLH-30 is also required for the increase in lifespan upon calcineurin inhibition. Because autophagy may regulate lifespan via multiple longevity pathways (*Hansen et al., 2008*; *Lapierre et al., 2013*), this could explain why calcineurin inhibition enhances lifespan via multiple mechanisms.

Dietary restriction-mediated lifespan extensions are complex and context-dependent (*Greer and Brunet, 2009*). Multiple longevity pathways have been identified under different paradigms of dietary restriction (*Chamoli et al., 2020*; *Chamoli et al., 2014*; *Chen et al., 2009*; *Greer and Brunet, 2009*; *Lapierre et al., 2013*; *Matai et al., 2019*; *Selman et al., 2009*; *Thondamal et al., 2014*). We found that *tax-6* knockdown enhanced lifespan independent of multiple dietary restriction pathways, including *raga-1*, *rsks-1*, *daf-16*, and *nhr-49*. An earlier study identified that *tax-6* knockdown enhanced lifespan via NHR-49 (*Burkewitz et al., 2015*). We currently do not understand the reasons for this discrepancy. We identified that *tax-6* knockdown resulted in the upregulation of *nhr-8* mRNA levels, and *nhr-8* was required for the increased lifespan in *tax-6* knockdown animals. NHR-8 has been shown to mediate calorie restriction-dependent lifespan via the regulation of xenobiotic responses (*Chamoli et al., 2014*; *Verma et al., 2018*). It remains to be studied whether NHR-8 and HLH-30 work in the same or different pathways downstream of calcineurin inhibition (*Figure 6C*).

Recent studies in different organisms have shown that gut bloating has profound effects on food-seeking behaviors, immunity, and lifespan (*Duvall et al., 2019*; *Filipowicz et al., 2021*; *Kumar et al., 2019*; *Min et al., 2021*; *Singh and Aballay, 2019b*; *Singh and Aballay, 2019c*). Several *C. elegans* mutants with defects in the DMP and bloated intestinal lumens are known to have dampened nutrient absorption, leading to reduced lipid deposition in the gut, mimicking calorie restriction (*Sheng et al., 2015*). It is possible that some of the effects of intestinal bloating on the host physiology are mediated by calorie restriction. Indeed, the neuropeptide Y receptors, which control a diverse set of behaviors, including appetite, are activated by gut bloating (*Singh and Aballay, 2019c*). Calorie restriction is known to induce neuropeptide Y, which might trigger feeding behaviors (*Aveleira et al., 2015*; *de Rijke et al., 2005*; *Ferreira-Marques et al., 2016*). The complete characterization of the physiological changes downstream of gut bloating may provide broad insights into the effects of gut physiology on behaviors, immunity, and lifespan.

# Materials and methods

### Key resources table

| Reagent type (species) or resource | Designation | Source or reference | Identifiers | Additional information |
|---|---|---|---|---|
| Strain, strain background (*Escherichia coli*) | OP50 | *Caenorhabditis* Genetics Center (CGC) | OP50 | |
| Strain, strain background (*E. coli*) | HT115(DE3) | Source BioScience | HT115(DE3) | |
| Strain, strain background (*Pseudomonas aeruginosa*) | PA14 | Frederick M Ausubel laboratory | PA14 | |
| Strain, strain background (*P. aeruginosa*) | PA14-GFP | Frederick M Ausubel laboratory | PA14-GFP | |
| Strain, strain background (*Caenorhabditis elegans*) | N2 Bristol | CGC | N2 | |
| Strain, strain background (*C. elegans*) | tax-6(p675) | CGC | PR675 | |
| Strain, strain background (*C. elegans*) | tax-6(ok2065) | CGC | RB1667 | |
| Strain, strain background (*C. elegans*) | fer-1(b232) | CGC | HH142 | |
| Strain, strain background (*C. elegans*) | eat-2(ad465) | CGC | DA465 | |
| Strain, strain background (*C. elegans*) | aak-2(ok524) | CGC | RB754 | |
| Strain, strain background (*C. elegans*) | raga-1(ok386) | CGC | VC222 | |
| Strain, strain background (*C. elegans*) | rsks-1(ok1255) | CGC | RB1206 | |
| Strain, strain background (*C. elegans*) | daf-16(mu86) | CGC | CF1038 | |
| Strain, strain background (*C. elegans*) | nhr-49(nr2041) | CGC | STE68 | |
| Strain, strain background (*C. elegans*) | hlh-30(tm1978) | CGC | JIN1375 | |
| Strain, strain background (*C. elegans*) | nhr-8(ok186) | CGC | AE501 | |
| Strain, strain background (*C. elegans*) | daf-9(rh50) | CGC | RG1228 | |
| Strain, strain background (*C. elegans*) | pmk-1(km25) | CGC | KU25 | |

*Continued on next page*

*Continued*

| Reagent type (species) or resource | Designation | Source or reference | Identifiers | Additional information |
|---|---|---|---|---|
| Strain, strain background (*C. elegans*) | *kgb-1(km21)* | CGC | KU21 | |
| Strain, strain background (*C. elegans*) | *dbl-1(nk3)* | CGC | NU3 | |
| Strain, strain background (*C. elegans*) | *zip-2(ok3730)* | CGC | VC3056 | |
| Strain, strain background (*C. elegans*) | *clk-1(qm30)* | CGC | MQ130 | |
| Strain, strain background (*C. elegans*) | *isp-1(qm150)* | CGC | MQ887 | |
| Strain, strain background (*C. elegans*) | *sid-1(qt9)* | CGC | HC196 | |
| Strain, strain background (*C. elegans*) | *eat-2(ad465);tax-6(ok2065)* | This study | | Materials and methods section |
| Sequence-based reagent | Pan-act_qPCR_F | This study | qPCR primers | TCGGTATGGGACAGAAGGAC |
| Sequence-based reagent | Pan-act_qPCR_R | This study | qPCR primers | CATCCCAGTTGGTGACGATA |
| Sequence-based reagent | pha-4_qPCR_F | This study | qPCR primers | CAAAGAGGAGCCAGAGTCGG |
| Sequence-based reagent | pha-4_qPCR_R | This study | qPCR primers | TGTTTCTGCTCGCGTTTTCG |
| Sequence-based reagent | nhr-8_qPCR_F | This study | qPCR primers | CTACACAGTTTCTCCGGCGT |
| Sequence-based reagent | nhr-8_qPCR_R | This study | qPCR primers | GCCATTTGGGCCATAACACC |
| Sequence-based reagent | tax-6(ok2065)_genotyping_F1 | This study | Genotyping primers | CTCCTTTGAGGGAGCCAGTG |
| Sequence-based reagent | tax-6(ok2065)_genotyping_F2 | This study | Genotyping primers | CTGGGGACAATCCACCATGAA |
| Sequence-based reagent | tax-6(ok2065)_genotyping_R1 | This study | Genotyping primers | TGTGTCCTGTATCTGTGGGC |
| Sequence-based reagent | eat-2(ad465)_genotyping_F | This study | Genotyping primers | CGGTGCAAAGAGCACATCTC |
| Sequence-based reagent | eat-2(ad465)_genotyping_R | This study | Genotyping primers | TTAAGGCGTACGAGCCTTCC |
| Software, algorithm | GraphPad Prism 8 | GraphPad Software | RRID:SCR_002798 | https://www.graphpad.com/scientificsoftware/prism/ |
| Software, algorithm | Photoshop CS5 | Adobe | RRID:SCR_014199 | https://www.adobe.com/products/photoshop.html |
| Software, algorithm | ImageJ | NIH | RRID:SCR_003070 | https://imagej.nih.gov/ij/ |

## Bacterial strains

The following bacterial strains were used: *Escherichia coli* OP50, *E. coli* HT115(DE3), *Pseudomonas aeruginosa* PA14, and *P. aeruginosa* PA14 expressing green fluorescent protein (*P. aeruginosa* PA14-GFP). The cultures for these bacteria were grown in Luria-Bertani (LB) broth at 37 °C.

### *C. elegans* strains and growth conditions

*C. elegans* hermaphrodites were maintained on nematode growth medium (NGM) plates seeded with *E. coli* OP50 at 20 °C unless otherwise indicated. Bristol N2 hermaphrodites were used as the wild-type control unless otherwise indicated. The strains used in the study are provided in the Key Resources Table. Some of the strains were obtained from the *Caenorhabditis* Genetics Center (University of Minnesota, Minneapolis, MN). The *eat-2(ad465);tax-6(ok2065)* strain was generated using a standard genetic cross. The *fer-1(b232)* hermaphrodites were maintained on *E. coli* OP50 at 15 °C and were grown at 25 °C prior to *P. aeruginosa* killing assays and longevity assays to obtain animals with unfertilized oocytes.

### RNA interference (RNAi)

RNAi was used to generate loss-of-function phenotypes by feeding nematodes *E. coli* strain HT115(DE3) expressing double-stranded RNA homologous to a target gene. RNAi was carried out as described previously (*Gokul and Singh, 2022*; *Ravi et al., 2023*). Briefly, *E. coli* with the appropriate vectors were grown in LB broth containing ampicillin (100 µg/mL) at 37 °C overnight and plated onto NGM plates containing 100 µg/mL ampicillin and 3 mM isopropyl β-D-thiogalactoside (IPTG) (RNAi plates). RNAi-expressing bacteria were allowed to grow overnight at 37 °C. Gravid adults were transferred to RNAi-expressing bacterial lawns and allowed to lay eggs for 2 hr. The gravid adults were removed, and the eggs were allowed to develop at 20 °C to 1-day-old adults for subsequent assays. The RNAi clones were from the Ahringer RNAi library and were verified by sequencing.

### RNAi efficiency test

The RNAi efficiency test was carried out as described earlier (*Gahlot and Singh, 2024*; *Ghosh and Singh, 2024*). Briefly, the wild-type N2, *eat-2(ad456), nhr-8(ok186)*, and *hlh-30(tm1978)* worms were synchronized by egg laying and grown on empty vector control, *act-5*, and *bli-3* RNAi plates at 20 °C. The *sid-1(qt9)* worms were used as RNAi-defective controls. After 72 hr, the animals were monitored for development defects and blisters on the cuticle for *act-5* and *bli-3* RNAi, respectively.

### *C. elegans* longevity assays

Lifespan assays were performed on RNAi plates containing *E. coli* HT115(DE3) with the empty vector control and *tax-6* RNAi clone in the presence of 50 µg/mL of 5-fluorodeoxyuridine (FUdR). Animals were synchronized on RNAi plates without FUdR and incubated at 20 °C. At the late L4 larval stage, the animals were transferred onto the corresponding RNAi plates containing 50 µg/mL of FUdR and incubated at 20 °C. For *fer-1(b232)* lifespan assays, the animals were synchronized on RNAi plates without FUdR and incubated at 25 °C. The *fer-1(b232)* lifespan assays were carried out at 25 °C without FUdR. Animals were scored every other day as live, dead, or gone. Animals that failed to display touch-provoked movement were scored as dead. Animals that crawled off the plates were censored. Young adult animals were considered as day 0 for the lifespan analysis. Three independent experiments were performed. The exact number of animals per condition per replicate is provided in the source data files.

### *C. elegans* killing assays on *P. aeruginosa* PA14

Bacterial cultures were prepared by inoculating individual bacterial colonies of *P. aeruginosa* into 2 mL of LB and growing them for 10–12 hr on a shaker at 37 °C. Bacterial lawns were prepared by spreading 20 µL of the culture on the entire surface of 3.5-cm-diameter modified NGM agar plates (0.35% instead of 0.25% peptone). The plates were incubated at 37 °C for 12–16 hr and then cooled to room temperature for at least 30 min before seeding with synchronized 1-day-old adult animals. The killing assays were performed at 25 °C, and live animals were transferred daily to fresh plates. Animals were scored at times indicated and were considered dead when they failed to respond to touch. The exact number of animals per condition per replicate is provided in the source data files.

### *P. aeruginosa*-GFP colonization assay

Bacterial cultures were prepared by inoculating individual bacterial colonies of *P. aeruginosa*-GFP into 2 mL of LB and growing them for 10–12 hr on a shaker at 37 °C. Bacterial lawns were prepared by spreading 20 µL of the culture on the entire surface of 3.5-cm-diameter modified NGM agar plates

(0.35% instead of 0.25% peptone) containing 50 µg/mL of kanamycin. The plates were incubated at 37 °C for 12 hr and then cooled to room temperature for at least 30 min before seeding with 1-day-old adult gravid adults. The assays were performed at 25 °C. After 6 hr of incubation, the animals were transferred from *P. aeruginosa*-GFP plates to fresh *E. coli* OP50 plates and visualized within 5 min under a fluorescence microscope.

## Quantification of intestinal bacterial loads

The intestinal bacterial loads were quantified by measuring colony-forming units (CFU) as described earlier (*Singh and Aballay, 2019c*; *Singh and Aballay, 2019b*) with appropriate modifications. Briefly, *P. aeruginosa*-GFP lawns were prepared as described above. After 6 hr of exposure, the animals were transferred from *P. aeruginosa*-GFP plates to the center of fresh *E. coli* OP50 plates thrice for 10 min each to eliminate bacteria stuck to their body. Afterward, 10 animals/condition were transferred into 50 µL of PBS containing 0.01% triton X-100 and ground using glass beads. Serial dilutions of the lysates ($10^1$, $10^2$, $10^3$, $10^4$) were seeded onto LB plates containing 50 µg/mL of kanamycin to select for *P. aeruginosa*-GFP cells and grown overnight at 37 °C. Single colonies were counted the next day and represented as the number of bacterial cells or CFU per animal. At least three independent experiments were performed for each condition.

## Fluorescence imaging

Fluorescence imaging was carried out as described previously (*Gokul and Singh, 2022*; *Ravi et al., 2023*). Briefly, the animals were anesthetized using an M9 salt solution containing 50 mM sodium azide and mounted onto 2% agarose pads. The animals were then visualized using a Nikon SMZ-1000 fluorescence stereomicroscope.

## Quantification of intestinal lumen bloating

The 1-day-old adult animals grown on the empty vector control and *tax-6* RNAi were anesthetized using an M9 salt solution containing 50 mM sodium azide, mounted onto 2% agarose pads, and imaged. The diameter of the intestinal lumen was measured using the Zeiss Zen Pro 2011 software. At least ten animals were used for each condition.

## Measurement of defecation motor program (DMP) rate

Wild-type N2 and *tax-6(p675)* animals were synchronized and grown at 20 °C on *E. coli* OP50 for 4 and 5 days, respectively, before measuring the DMP rate. For RNAi experiments, N2 worms were synchronized on EV and *tax-6* RNAi plates and grown for 4 days at 20 °C before measuring the DMP rate. The DMP cycle length was scored by assessing the time between expulsions (which are preceded by posterior and anterior body wall muscle contraction and the contraction of enteric muscles in a normal regular pattern; *Thomas, 1990*). The number of expulsion events in 20 min was measured for each worm. The DMP rate was recorded for 5–6 worms/condition.

## Ethogram for DMP

Wild type N2 and *tax-6(ok2065)* worms were synchronized by egg laying and grown at 20 °C on *E. coli* OP50 for 4 and 5 days, respectively. For RNAi experiments, N2 worms were synchronized and grown on empty vector control and *tax-6* RNAi at 20 °C for 4 days. The assay was performed at 25 °C. The defecation motor program consists of three steps- posterior body-wall muscle contraction (p), anterior body-wall muscle contraction (a), and expulsion muscle contraction (x). Animals were observed for the exact time of these three contractions on a dissecting scope for a continuous period of 20 min. A computer program was developed to plot the ethogram where we manually input the timings of 'p', 'a', and 'x' contractions.

## Blue dye assay to assess functional defecation rate

Wild type N2 and *tax-6(ok2065)* worms were synchronized by egg laying and grown at 20 °C on *E. coli* OP50 for 4 and 5 days, respectively, before measuring the functional defecation rate. For RNAi experiments, N2 worms were synchronized and grown on empty vector control and *tax-6* RNAi at 20 °C for 4 days. The functional defecation rate was assessed using a blue dye, erioglaucine disodium salt (5% wt/v), as an indicator of gut content. The time required by worms to clear the blue dye from

the gut was used to calculate a functional defecation rate. Briefly, 50 worms from each condition were incubated in a solution containing blue dye and *E. coli* OP50 culture in a 1:1 ratio for 2 hr. Then, the worms were transferred to an NGM plate seeded with *E. coli* OP50. About 25–27 worms from each condition were transferred to a fresh NGM plate seeded with *E. coli* OP50. The functional defecation rate was assessed by observing the clearance of blue dye from the intestine of worms every 5 min over a period of 60 min. Three independent experiments were performed for each condition.

## Pharyngeal pumping assay

For the pharyngeal pumping assay, 1-day-old adult animals grown on the empty vector control and *tax-6* RNAi were used. The number of contractions of the terminal bulb of the pharynx was counted for 30 s per worm. The pumping rates for 20 worms were recorded for each condition.

## Lipid staining

The oil-red-O (ORO) staining was carried out as described earlier (*Lynn et al., 2015*) with appropriate modification. Briefly, wild-type N2, *hlh-30(tm1978)*, and *nhr-8(ok186)* animals were synchronized on empty vector control and *tax-6* RNAi bacteria and grown at 20 °C for 4 days. About 300–400 synchronized gravid adult worms from experimental plates were washed three times with 1 X PBS plus 0.01% triton X-100 (PBST). To permeabilize the cuticle, 600 µL of 40% isopropanol was added to 100 µL of worm pellet and was rocked for 3 min. The animals were spun down at 500 rpm for 30 s, and then 600 µL of supernatant was removed. Subsequently, 600 µL of ORO working stock was added to the worm pellet to stain the worms and incubated at room temperature for 1 hr on a shaker. ORO working stock was freshly prepared by diluting the stock 0.5% ORO in isopropanol to 60% in water and rocked at room temperature for 2 hr, followed by the removal of debris with a 0.22 µm filter. Afterward, worm samples were pelleted, 600 µL of supernatant was removed, 600 µL of PBST was added, and the samples were rocked for 30 min at room temperature. After that, the worms were mounted on a 2% agarose pad and imaged. At least 10 worms/condition were imaged. Three independent biological replicates were performed. The ORO intensity was quantified using Image J software.

## RNA isolation and quantitative reverse transcription-PCR (qRT-PCR)

Animals were synchronized by egg laying. Approximately 40 N2 gravid adult animals were transferred to 9 cm RNAi plates seeded with *E. coli* HT115 expressing the appropriate vectors and allowed to lay eggs for 4–5 hr. The gravid adults were then removed, and the eggs were allowed to develop at 20 °C for 96 hr. Subsequently, the animals were collected, washed with M9 buffer, and frozen in TRIzol reagent (Life Technologies, Carlsbad, CA). Total RNA was extracted using the RNeasy Plus Universal Kit (QIAGEN, Netherlands). Residual genomic DNA was removed using TURBO DNase (Life Technologies, Carlsbad, CA). A total of 6 µg of total RNA was reverse-transcribed with random primers using the High-Capacity cDNA Reverse Transcription Kit (Applied Biosystems, Foster City, CA).

qRT-PCR was conducted using the Applied Biosystems One-Step Real-time PCR protocol using SYBR Green fluorescence (Applied Biosystems) on an Applied Biosystems 7900HT real-time PCR machine in 96-well-plate format. Twenty-five-microliter reactions were analyzed as outlined by the manufacturer (Applied Biosystems). The relative fold-changes of the transcripts were calculated using the comparative $CT(2^{-\Delta\Delta CT})$ method and normalized to pan-actin (*act-1,–3, –4*) as described earlier (*Singh and Aballay, 2019b*; *Singh and Aballay, 2017*). The cycle thresholds of the amplification were determined using StepOnePlus software (Applied Biosystems). All samples were run in triplicate. The primer sequences are provided in the Key Resources Table.

## Quantification and statistical analysis

The statistical analysis was performed with Prism 8 (GraphPad). All error bars represent mean ± standard deviation (SD). The unpaired, two-tailed, two-sample *t*-test was used when needed, and the data were judged to be statistically significant when $p < 0.05$. In the figures, asterisks (*) denote statistical significance as follows: *, $p < 0.05$, **, $p < 0.01$, ***, $p < 0.001$, as compared with the appropriate controls. The Kaplan-Meier method was used to calculate the survival fractions, and statistical significance between survival curves was determined using the log-rank test. All experiments were performed in triplicate. The exact number of animals per condition per replicate is provided in the source data files.

## Acknowledgements

We thank Sagar Santosh Gawande for help in plotting the DMP ethogram. Some strains used in this study were provided by the *Caenorhabditis* Genetics Center (CGC), which is funded by the NIH Office of Research Infrastructure Programs (P40 OD010440). Funding: This work was supported by the following grants: Har-Gobind Khorana-Innovative Young Biotechnologist Fellowship (File No. HRD-17011/2/2023-HRD-DBT) and Ramalingaswami Re-entry Fellowship (Ref. No. BT/RLF/Re-entry/50/2020) awarded by the Department of Biotechnology, India; STARS grant (File No. MoE-STARS/STARS-2/2023–0116) awarded by the Ministry of Education, India; Research Grant (Ref. No. 37/1741/23/EMR-II) awarded by the Council of Scientific & Industrial Research (CSIR), India; Science and Engineering Research Board (SERB) Startup Research Grant (Ref. No. SRG/2020/000022) and Core Research Grant (Ref. No. CRG/2023/001136) awarded by DST, India; and IISER Mohali intramural funds. PD was supported by a senior research fellowship from the Council of Scientific & Industrial Research (CSIR), India.

## Additional information

### Funding

| Funder | Grant reference number | Author |
| --- | --- | --- |
| Department of Biotechnology, Ministry of Science and Technology, India | HRD-17011/2/2023-HRD-DBT | Jogender Singh |
| Department of Biotechnology, Ministry of Science and Technology, India | BT/RLF/Re-entry/50/2020 | Jogender Singh |
| Ministry of Education, India | MoE-STARS/STARS-2/2023-0116 | Jogender Singh |
| Council of Scientific and Industrial Research, India | 37/1741/23/EMR-II | Jogender Singh |
| Science and Engineering Research Board | CRG/2023/001136 | Jogender Singh |
| Science and Engineering Research Board | SRG/2020/000022 | Jogender Singh |

The funders had no role in study design, data collection and interpretation, or the decision to submit the work for publication.

### Author contributions

Priyanka Das, Conceptualization, Data curation, Formal analysis, Investigation, Methodology; Alejandro Aballay, Conceptualization, Supervision, Visualization; Jogender Singh, Conceptualization, Data curation, Formal analysis, Supervision, Funding acquisition, Investigation, Visualization, Methodology, Writing – original draft, Project administration, Writing – review and editing

### Author ORCIDs

Alejandro Aballay ⬛ https://orcid.org/0000-0002-5975-3352
Jogender Singh ⬛ https://orcid.org/0000-0002-7947-0405

Reviewer #1 (Public review): https://doi.org/10.7554/eLife.89572.3.sa1
Reviewer #2 (Public review): https://doi.org/10.7554/eLife.89572.3.sa2
Author response https://doi.org/10.7554/eLife.89572.3.sa3

# Additional files

## Supplementary files
• MDAR checklist

## Data availability
All data generated or analyzed during this study are included in the manuscript and supporting files; source data files have been provided for Figures 1, 2, 3, 4, 5, 6, Figure 1—figure supplement 1 and 2, Figure 2—figure supplement 1 and 2, Figure 3—figure supplement 3, and Figure 5—figure supplement 2.

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
