## [Editor Report · eLife Assessment]

This **important** study reveals insights into how calcineurin influences *C. elegans* pathogen susceptibility and lifespan through its role in controlling the defecation motor program. The authors provide **convincing** evidence to support a new mechanism through which calcineurin impacts longevity. This work will be of interest to investigators studying host-pathogen interactions and aging in a number of experimental systems.

---

## [Referee Report · Reviewer #1 (Public review)]

In this paper, the authors show that disruption of calcineurin, which is encoded by tax-6 in *C. elegans*, results in increased susceptibility to *P. aeruginosa* but extends lifespan. In exploring the mechanisms involved, the authors show that disruption of tax-6 decreases the rate of defecation leading to intestinal accumulation of bacteria and distension of the intestinal lumen. The authors further show that the lifespan extension is dependent on hlh-30, which may be involved in breaking down lipids following deficits in defecation, and nhr-8, whose levels are increased by deficits in defecation. The authors propose a model in which disruption of the defecation motor program is responsible for the effect of calcineurin on pathogen susceptibility and lifespan, but do not exclude the possibility that calcineurin affects these phenotypes independently of defecation.

---

## [Referee Report · Reviewer #2 (Public review)]

The relationships between genes and phenotypes are complex and the impact of deleting or a gene can often have multifaceted and unforeseen consequences. This paper dissected the role of calcineurin, encoded by tax-6, in various phenotypes in *C. elegans*, including lifespan, pathogen susceptibility, the defecation motor program, and nutrient absorption or calorie restriction, through a series of genetic and behavioral analyses. Many genes in these pathways were tested yielding robust results. Classic epistasis analyses were used to distinguish between genes operating in the same or separate pathways. Researchers in the related fields will be very interested in looking through the data presented in this paper in great detail and benefit from it.

Overall, this paper supports a model in which the increased lifespan and heightened pathogen susceptibility observed following calcineurin inhibition result from the disruptions in the defecation motor program but through distinct pathways. A defective defecation motor program leads to intestine bloating and compromised nutrient absorption. Calorie restriction resulting from poor nutrient absorption affects lifespan, whereas increased colonization in the bloated intestine heightens pathogen susceptibility. The observation that knockdown of several other DMP-related genes also results in increased lifespan and pathogen susceptibility further reinforces the proposed model.

---

## [Author Response]

The following is the authors’ response to the original reviews.

We would like to thank the reviewers for their interest in our studies. In response to their comments, we have conducted additional experiments and made the necessary revisions to the manuscript. The new studies included to address the reviewers’ comments are shown in Figure 1B, 1F, Figure 2—figure supplement 1, Figure 3, Figure 3—figure supplement 1, Figure 3—figure supplement 2, Figure 3—figure supplement 3, Figure 4E, Figure 4—figure supplement 1, Figure 5, Figure 5—figure supplement 1, Figure 5—figure supplement 2D, and Figure 6. We are grateful for the critiques, which have helped us substantially improve the quality of the manuscript.

Below, we have provided a point-by-point response to the reviewers’ comments.

**Public Reviews:**

**Reviewer #1 (Public Review):**
In this paper, the authors show that disruption of calcineurin, which is encoded by tax-6 in *C. elegans*, results in increased susceptibility to *P. aeruginosa*, but extends lifespan. In exploring the mechanisms involved, the authors show that disruption of tax-6 decreases the rate of defecation leading to intestinal accumulation of bacteria and distension of the intestinal lumen. The authors further show that the lifespan extension is dependent on hlh-30, which may be involved in breaking down lipids following deficits in defecation, and nhr-8, whose levels are increased by deficits in defecation. The authors propose a model in which disruption of the defecation motor program is responsible for the effect of calcineurin on pathogen susceptibility and lifespan, but do not exclude the possibility that calcineurin affects these phenotypes independently of defecation.

We thank the reviewer for providing an excellent summary of our work. We have performed additional experiments as suggested by both the reviewers and believe we have thoroughly addressed all the reviewers' concerns.

**Reviewer #2 (Public Review):**
The manuscript titled "Calcineurin Inhibition Enhances *Caenorhabditis elegans* Lifespan by Defecation Defects-Mediated Calorie Restriction and Nuclear Hormone Signaling" by Priyanka Das, Alejandro Aballay, and Jogender Singh reveals that inhibiting calcineurin, a conserved protein phosphatase, in *C. elegans* affects the defecation motor program (DMP), leading to intestinal bloating and increased susceptibility to bacterial infection. This intestinal bloating mimics calorie restriction, ultimately resulting in an enhanced lifespan. The research identifies the involvement of HLH-30 and NHR-8 proteins in this lifespan enhancement, providing new insights into the role of calcineurin in *C. elegans* DMP and mechanisms for longevity.The authors present novel findings on the role of calcineurin in regulating the defecation motor program in *C. elegans* and how its inhibition can lead to lifespan enhancement. The evidence provided is solid with multiple experiments supporting the main claims.Strengths:The manuscript's strength lies in the authors' use of genetic and biochemical techniques to investigate the role of calcineurin in regulating the DMP, innate immunity, and lifespan in *C. elegans*. Moreover, the authors' findings provide a new mechanism for calcineurin inhibitionmediated longevity extension, which could have significant implications for understanding the molecular basis of aging and developing interventions to promote healthy aging.(1) The study uncovers a new role for calcineurin in the regulation of *C. elegans* DMP and a potential novel pathway for enhancing lifespan via calorie restriction involving calcineurin, HLH-30, and NHR-8 in *C. elegans*.(2) Multiple signaling pathways involved in lifespan enhancement were investigated with fairly strong experimental evidence supporting their claims.

We thank the reviewer for an excellent summary of our work and for highlighting the strengths of the findings.

Weaknesses:The manuscript's weaknesses include the lack of mechanistic details regarding how calcineurin inhibition leads to defects in the DMP and induces calorie restriction-like effects on lifespan.The exact site of calcineurin action, i.e., whether in the intestine or enteric muscles (Lee et al., 2005), and the possible molecular mechanisms linking calcineurin inhibition, DMP defects, and lifespan were not adequately explored. Although characterization of the full mechanism is probably beyond the scope of this paper, given the relative simplicity and advantages of using *C. elegans* as a model organism for this study, some degree of rigor is expected with additional straightforward control experiments as listed below:The authors state that tax-6 knockdown animals had drastically reduced expulsion events (Figure 2G), leading to irregular DMP (Lines 144-145), but did not describe the nature of DMP irregularity. For example, did the reduced expulsion events still occur with regular intervals but longer cycle lengths? Or was the rhythmicity completely abolished? The former would suggest the intestine clock is still intact, and the latter would indicate that calcineurin is required for the clock to function. Therefore, ethograms of DMP in both wild-type and tax6 mutant animals are warranted to be included in the manuscript. Along the same line, besides the cycle length, the three separable motor steps (aBoc, pBoc, EMC) are easily measurable, with each step indicating where the program goes wrong, hence the site of action, which is precisely the beauty of studying *C. elegans* DMP. Unfortunately, the authors did not use this opportunity to characterize the exact behavior phenotypes of the tax-6 mutant to guide future investigations. Furthermore, it is interesting that about 64% of tax-6 (p675) animals had normal DMP. The authors attributed this to p675 being a weak allele. It would be informative to further examine tax-6 RNAi as in other experiments or to make a tax-6 null mutant with CRISPR. In addition, in one of the cited papers (Lee et al., 2005), the exact calcineurin loss-of-function strain tax-6(p675) was shown to have normal defecation, including normal EMC, while the gain-of-function mutant of calcineurin tax-6(jh107) had abnormal EMC steps. It wasn't clear from Lee et al., 2005, if the reported "normal defecation" was only referring to the expulsion step or also included the cycle length. Nevertheless, this potential contradiction and calcineurin gain-of-function mutant is highly relevant to the current study and should be further explored as a follow-up to previously reported results. For some of the key experiments, such as tax-6's effects on susceptibility to PA14, DMP, intestinal bloating, and lifespan, additional controls, as the norm of *C. elegans* studies, including second allele and rescue experiments, would strengthen the authors' claims and conclusions.

We have now included lifespan, survival on *P. aeruginosa*, and DMP data using an additional knockout allele, tax-6(ok2065). Additionally, we have added ethograms of DMP for both tax-6 RNAi and the tax-6(ok2065) mutant. Our observations indicate that tax-6 inhibition leads to a complete loss of DMP rhythmicity, suggesting that calcineurin is essential for maintaining the DMP clock. While characterizing the DMP, we noticed that expulsion events appeared superficial in the tax-6(ok2065) mutant, with little to no gut content released. Consequently, we examined the movement of gut content and found that both tax-6(ok2065) mutants and tax-6 knockdown animals showed significantly reduced gut content movement. The new findings on the characterization of DMP are presented in Figure 2—figure supplement 1, Figure 3, Figure 3—figure supplement 1, and Figure 3—figure supplement 2. The text in the results section reads (lines 160-176): “Next, we investigated whether the reduced number of expulsion events was due to regular intervals with longer cycle lengths or if rhythmicity was entirely disrupted upon tax-6 knockdown. To assess this, we obtained ethograms of the DMP for N2 animals grown on control and tax-6 RNAi. While animals on control RNAi displayed regular cycles of pBoc, aBoc, and EMC, the tax-6 RNAi animals exhibited disrupted rhythmicity (Figure 3A and Figure 3—figure supplement 1). Most tax-6 knockdown animals lacked the pBoc and aBoc steps and had sporadic expulsion events. Isolated pBoc events were occasionally observed, indicating a complete loss of rhythmicity in tax-6 knockdown animals. Ethograms for tax-6(ok2065) animals also showed disrupted rhythmicity (Figure 3B and Figure 3—figure supplement 2). Although the number of expulsion events appeared higher in tax-6(ok2065) animals compared to tax-6 RNAi animals (Figure 3—figure supplement 1 and 2), these expulsion events seemed superficial, releasing little to no gut content. This suggested slow movement of gut content in tax6(ok2065) animals, leading to constipation and intestinal bloating. We examined gut content movement by measuring the clearance of blue dye (erioglaucine disodium salt) from the gut. The clearance was significantly slower in tax-6(ok2065) animals compared to N2 animals (Figure 3C), indicating impaired gut content movement due to the loss of tax-6. Similarly, tax-6 knockdown animals also showed significantly slowed gut content movement (Figure 3D).”

Moreover, we have added a potential reason for the tax-6(p675) contradictory results from Lee et al., 2005 (lines 154-159): “At the 1-day-old adult stage, about 36% of tax-6(p675) animals showed irregular and slowed DMP, while the remainder had regular DMP (Figure 2H), suggesting that tax-6(p675) is a weak allele. The fraction of the animals with irregular DMP appeared to increase with age, indicating that this phenotype might be agedependent. This may also explain why tax-6(p675) animals were reported to have a normal defecation cycle in an earlier study (Lee et al., 2005).”

The second weakness of this manuscript is the data presentation for all survival rate curves. The authors stated that three independent experiments or biological replicates were performed for each group but only showed one "representative" curve for each plot. Without seeing all individual datasets or the averaged data with error bars, there is no way to evaluate the variability and consistency of the survival rate reported in this study.

We now provide all replicates data in the source data files.

Overall, the authors' claims and conclusions are justified by their data, but further experiments are needed to confirm their findings and establish the detailed mechanisms underlying the observed effects of calcineurin inhibition on the DMP, calorie restriction, and lifespan in *C. elegans*.

We have conducted additional experiments to elucidate the role of calcineurin in the DMP and to investigate the impact of the DMP on calorie restriction and lifespan in *C. elegans*, as described in the various responses to the reviewers’ comments.

**Recommendations for the authors:**
Our specific comments to guide the authors, should they choose to revise the manuscript:The RNAi experiments in the eat-2 mutant background are difficult to interpret. If these animals are eating fewer bacteria, it is possible that there is also less tax-6 dsRNA being ingested and therefore less tax-6 inactivation. These experiments should be conducted with a tax-6 null allele.

We have included lifespan experiments with the eat-2(ad465);tax-6(ok2065) double mutant, along with the individual single mutant controls, as shown in Figure 4E. These results demonstrate that the eat-2 mutation does not further extend the lifespan of the tax-6(ok2065) mutant. Additionally, we confirmed that the eat-2(ad465) mutants do not exhibit defects in feeding-based RNAi (Figure 4—figure supplement 1).

While aak-2, hlh-30, and nhr-8 mutants may not have an eat phenotype, the negative tax-6 RNAi results should be confirmed with a tax-6 null mutant to obviate the consideration that these background mutations reduce RNAi efficacy.

The genes hlh-30 and nhr-8 are located very close to tax-6 on chromosome IV (https://wormbase.org//#012-34-5), which made it challenging to generate double mutants. However, we tested the RNAi sensitivity of the hlh-30(tm1978) and nhr-8(ok186) mutants and confirmed that they are not defective in RNAi (Figure 5—figure supplement 1). We also found that tax-6 RNAi disrupted the DMP in both hlh-30(tm1978) and nhr-8(ok186) mutants (Figure 5—figure supplement 2). Furthermore, our results show that hlh-30(tm1978) and nhr-8(ok186) animals have increased susceptibility to *P. aeruginosa* upon tax-6 knockdown (Figure 6A, B), indicating that tax-6 RNAi was effective in these mutants. Since the phenotype in the aak-2 mutant was only partially observed, we did not conduct further experiments with aak-2 mutants.

**Reviewer #1 (Recommendations For The Authors):**
The low penetrance of defecation cycle defects in tax-6(p675) worms brings into question the role of the defecation deficits in the phenotypes caused by the disruption of tax6. At the same time, the low penetrance provides a golden opportunity to test this. Do tax6(p675) worms with a normal defecation cycle length have extended longevity? Increased susceptibility to bacterial pathogens? Smaller body size? Distended lumen? Decreased fat accumulation? Increased pha-4 and nhr-8 expression? It would be relatively straightforward to measure defecation cycle length in individual tax-6(p675) worms, bin them into normal defecation and slow defecation groups, and then compare the above-mentioned phenotypes.

We appreciate the reviewer's interesting suggestion. However, the DMP defect phenotype in tax-6(p675) worms appears to be age-dependent, with the number of DMPdefective worms increasing as they age. Additionally, we observed that exposure to *P. aeruginosa* accelerates the onset of DMP defects in tax-6(p675) worms. As a result, tax6(p675) worms are not suitable for the type of experiments the reviewer suggested. Nevertheless, we believe that the additional data using the tax-6(ok2065) mutant, along with the characterization of ethograms of DMP, firmly establishes the role of calcineurin in maintaining a regular DMP in *C. elegans*.

Another way to dissect specific effects of calcineurin disruption from phenotypes resulting from defecation motor program deficits would be to further characterize other worms with deficits in defecation (flr-1, nhx-2, pbo-1 RNAi). It is mentioned that they have decreased lifespan. Do they also show increased susceptibility to bacterial pathogens? Do they show decreased fat? Is their lifespan dependent on HLH-30 and NHR-8?

We thank the reviewer for this important suggestion. We have now included data with flr-1, nhx-2, and pbo-1 RNAi, which shows that the knockdown of these genes also enhances susceptibility to *P. aeruginosa* (Figure 3—figure supplement 3G). Knockdown of these genes is already known to reduce fat levels in N2 worms, and we demonstrate that they similarly reduce fat levels in hlh-30(tm1978) and nhr-8(ok186) animals (Figure 5B, C, F, G). Additionally, we found that the increased lifespan observed upon knockdown of these genes (as well as with tax-6 knockdown) is dependent on HLH-30 and NHR-8 (Figure 5A, D).

To place "enhanced susceptibility to pathogen" within the proposed model, it would be important to examine the effect of HLH-30 and NHR-8 disruption on this phenotype. The proposed model suggests that this phenotype is independent of HLH-30 and NHR-8, but this should be tested experimentally. Similarly, it would be important to test the effect of HLH-30 and NHR-8 disruption on defecation cycle length to determine if defecation deficits are upstream or downstream of deficits in the defecation motor program

We show that the knockdown of tax-6 leads to defects in the DMP in hlh30(tm1978) and nhr-8(ok186) animals (Figure 5—figure supplement 2). Moreover, we show that hlh-30(tm1978) and nhr-8(ok186) animals have increased susceptibility to *P. aeruginosa* upon tax-6 knockdown (Figure 6A, B). These results are described as (lines 279-285): “Given that HLH-30 and NHR-8 are essential for lifespan extension upon calcineurin inhibition, we investigated whether these pathways also influence survival in response to P. aeruginosa infection following calcineurin knockdown. Both hlh-30(tm1978) and nhr-8(ok186) animals showed significantly reduced survival upon tax-6 RNAi (Figure 6A, B). These findings suggested that the reduced survival on *P. aeruginosa* following calcineurin inhibition is independent of HLH-30 and NHR-8 and is more likely due to increased gut colonization by *P. aeruginosa* resulting from DMP defects (Figure 6C).”

Is the lifespan of tax-6(p675) increased? This would be important to measure and include in Figure 1.

Indeed, the lifespan of tax-6(p675) mutants is increased. We have included the lifespan of tax-6(p675) and tax-6(ok2065) in Figure 1F.

In Figure 2, disruption of tax-6 appears to result in a clear decrease in body size. To what extent is the decrease in fat/worm in Figure 3 simply a result of the worms being smaller? Perhaps, a measurement of Oil-Red-O intensity PER AREA would be a more appropriate measure.

The ORO intensity values we had shown per animal were already area normalized. We have now indicated this in the Figure Legends.

There are multiple long-lived mutant strains such as clk-1 and isp-1 that have an increased defecation cycle length. To what extent do these worms exhibit phenotypes similar to tax-6 disruption? isp-1 have increased resistance to bacterial pathogens suggesting that defecation motor program deficits are not sufficient to increase susceptibility to bacterial pathogens.

We have now examined the clk-1 and isp-1 mutants and found that these mutants exhibit reduced gut colonization by *P. aeruginosa* compared to N2 animals. This reduction in colonization may be attributed to the slowed pharyngeal pumping rates observed in these mutants. These findings suggest that the phenotypes associated with a slow DMP versus a disrupted DMP could be significantly different. The manuscript with the new data on these mutants reads (lines 177-192): “We then explored whether the disruption of DMP rhythmicity due to tax-6 knockdown affected *P. aeruginosa* responses similarly to longer but regular DMP cycles. To do this, we studied *P. aeruginosa* colonization in clk-1(qm30) and isp1(qm150) mutants, which have regular but extended DMP cycles (Feng et al., 2001; Wong et al., 1995). Interestingly, both clk-1(qm30) and isp-1(qm150) mutants showed significantly reduced intestinal colonization by P. aeruginosa compared to N2 animals (Figure 3—figure supplement 3A-D). This reduced colonization could be attributed to their significantly decreased pharyngeal pumping rates (Wong et al., 1995; Yee et al., 2014), suggesting a lower intake of bacterial food in these mutants. While the survival of clk-1(qm30) animals on *P. aeruginosa* was comparable to N2 animals (Figure 3—figure supplement 3E), isp1(qm150) animals exhibited significantly improved survival (Figure 3—figure supplement 3F). Conversely, knockdown of flr-1, nhx-2, and pbo-1 in N2 animals resulted in significantly reduced survival on *P. aeruginosa* compared to control RNAi (Figure 3—figure supplement 3G). Knockdown of these genes causes complete disruption of DMP rhythmicity, increasing gut colonization by *P. aeruginosa* (Singh and Aballay, 2019a). Overall, these findings demonstrated that calcineurin is crucial for maintaining the DMP ultradian clock, and its inhibition increases susceptibility to *P. aeruginosa* by disrupting the DMP.”

Line 192. This statement is speculative. There is no evidence that HLH-30 is mediating lipid depletion in these worms.

We have removed this statement. We observed that the knockdown of flr-1, nhx2, and pbo-1 resulted in significant fat depletion in hlh-30(tm1978) animals (Figure 5B, C). Additionally, tax-6 knockdown also caused a small but significant reduction in fat levels in hlh-30(tm1978) animals. This contrasts with our initial submission, possibly due to the increased number of animals included in the analysis. These findings suggest that the increase in lifespan due to DMP defects requires HLH-30, likely through a mechanism independent of HLH-30’s role in fat depletion. We have updated the manuscript text and model (Figure 6C) accordingly.

In Figure S2, tax-6 RNAi appears to have a more detrimental effect in pmk-1 mutants than the other mutants. The authors should comment on this.

We have added the following sentence in the manuscript (lines 123-125): “The knockdown of tax-6 appeared to have a more pronounced effect in pmk-1(km25) mutants than in other mutants, suggesting that inhibition of tax-6 might exacerbate the adverse effects observed in pmk-1(km25) mutants.”

**Reviewer #2 (Recommendations For The Authors):**
Line 192-193: The statement is confusing and not accurate because HLH-30 did not enhance lifespan with or without calcineurin (Figure 4A and S4A, also in Lapierre 2023). The takeaway should be along the lines of calcineurin inhibition enhancing lifespan through HLH-30 or HLH-30 being required for lifespan enhancement via calcineurin inhibition.

We have removed this statement. We now state (lines 237-239): “Knockdown of tax-6 did not extend the lifespan of hlh-30(tm1978) animals (Figure 5A), indicating that HLH-30 is required for the increased lifespan observed with calcineurin inhibition.”

Line 261: Similar to the point above. Where is the data showing NHR-8 increases lifespan with or without calcineurin?

We have removed this sentence.

Figure 1 legend line 699: animals per condition per replicate >90, but in the Method section Line 317, it says more than 80 animals per condition per replicate. Could be more accurate.

We have now specified in the Methods section that the exact number of animals per condition is provided in the source data files. Since different lifespan curves within a given figure panel had varying numbers of animals, we have indicated the lower boundary for all curves (including the replicates). The precise number of animals for each lifespan experiment is available in the source data files.

Figures 2F and G, "tax-6" should be labeled as "tax-6 RNAi" to be consistent with other figures.

We thank the reviewer for this suggestion and have updated the label to “tax-6 RNAi”.

In summary, we would like to thank the reviewers again for providing constructive critiques. We believe we have fully addressed all the concerns of the reviewers by carrying out several new experiments and modifying the text. The manuscript has undergone substantial revision and has thereby improved significantly. We do hope that the evidence in support of the conclusions is found to be complete in the revised manuscript.